# Dietary salt with nitric oxide deficiency induces nocturnal polyuria in mice via hyperactivation of intrarenal angiotensin II-SPAK-NCC pathway

Y. Sekii [1], H. Kiuchi [1✉], K. Takezawa [1], T. Imanaka [1], S. Kuribayashi[1], K. Okada [1], Y. Inagaki [1], N. Ueda [1], S. Fukuhara [1], R. Imamura[1], H. Negoro[2] & N. Nonomura[1]

Nocturnal polyuria is the most frequent cause of nocturia, a common disease associated with a compromised quality of life and increased mortality. Its pathogenesis is complex, and the detailed underlying mechanism remains unknown. Herein, we report that concomitant intake of a high-salt diet and reduced nitric oxide (NO) production achieved through Nω-Nitro-L-arginine methyl ester hydrochloride (L-NAME) administration in mice resulted in nocturnal polyuria recapitulating the clinical features in humans. High salt intake under reduced NO production overactivated the angiotensin II-SPAK (STE20/SPS1-related proline–alanine-rich protein kinase)-NCC (sodium chloride co-transporter) pathway in the kidney, resulting in the insufficient excretion of sodium during the day and its excessive excretion at night. Excessive Na excretion at night in turn leads to nocturnal polyuria due to osmotic diuresis. Our study identified a central role for the intrarenal angiotensin II-SPAK-NCC pathway in the patho-physiology of nocturnal polyuria, highlighting its potential as a promising therapeutic target.

[1] Department of Urology, Graduate School of Medicine, Osaka University, Suita, Japan. [2] Department of Urology, Faculty of Medicine, University of Tsukuba, Tsukuba, Japan. ✉email: kiuchi@uro.med.osaka-u.ac.jp

Nocturia is characterized by the need to wake up to pass urine during the main sleep period[1]. It is a common and bothersome health problem in middle-aged and elderly individuals, affecting ~65% of all adults over the age of 50[2]. Notably, recent large-scale epidemiological research revealed an association between mortality rate and the frequency of urination at night: 1.5-fold increase for one urination vs. zero, 1.9-fold for two urinations, and 2.1-fold for three or more urinations[3]. A recent meta-analysis also indicated that nocturia was associated with a 1.2- to 1.3-fold increase in all-cause mortality[4,5]. Despite its high prevalence and association with mortality, the significance of nocturia has been overlooked for decades. There are multiple possible etiologies for nocturia, including decreased bladder capacity[6], sleep disturbances[7], nocturnal polyuria[8], and the combination of all three[3,6]. Among these etiologies, nocturnal polyuria accounts for 50–80% of nocturia cases[9], indicating that treatment of the former is crucial for the management of the latter. However, the efficacy of currently available clinical treatment for nocturnal polyuria such as behavioral therapy or desmopressin is limited[10–12]. This limited efficacy stems in part from multiple co-morbidities such as polydipsia[13], chronic kidney disease[14], and heart failure[15]. More importantly, treatment efficacy remains suboptimal due to a limited understanding of underlying pathophysiological mechanisms resulting from the lack of optimal preclinical models that reproduce the clinical features seen in nocturnal polyuria patients. Therefore, the generation of representative animal models and the development of novel therapeutic strategies for nocturnal polyuria are urgently needed.

High salt intake has been shown to increase the risk of nocturnal polyuria in middle-aged and older subjects[16,17]. Recently, we reported that a shift in salt excretion from daytime to nighttime causes an increase in nighttime urine volume[18]. These findings highlighted the implication of salt intake in the pathogenesis of nocturnal polyuria. However, despite salt intake being similar between generations[19,20], the frequency of nocturnal polyuria increases with age[21], suggesting the simultaneous involvement of additional age-related factors in the pathophysiology of nocturnal polyuria. We hypothesized that age-related changes affect salt excretion, leading to nocturnal polyuria, and aimed to establish a novel animal model of the condition by combining high salt intake and age-related factors in order to elucidate the underlying molecular mechanism.

## Results

### The effect of salt loading on diurnal polyuria differed between young and aged mice.
To assess whether aging influences the effects of salt loading on nocturnal urine production, we determined the salinity of the high-salt diet (HSD) used in animal experiments, as excess salt intake may itself increase urine volume. Generally, nocturnal polyuria does not exhibit polyuria, but a shift in urine volume from day to night. Therefore, we decided to set the diet salt level in so that urine volume would not differ between mice fed an HSD and a normal salt diet (NSD, 0.2% NaCl). We fed young mice with the NSD or 1%, 2%, and 4% HSD. We then measured the 24-h urine volume. No difference in daily urine volume was observed between mice fed the 1% HSD and those fed the NSD (Fig. 1a, 2220 vs. 2529 μl, n.s.). However, both the 2% and 4% HSD significantly increased urine volume (Fig. 1a, 2220 vs. 4614 μl and 2220 vs. 6087 μl, $p < 0.05$, respectively). Thus, we set the salt level to 1% for further studies. Next, we fed young (19 weeks old) and aged (80 weeks old) mice with NSD or 1% HSD for 2 weeks. The time and volume of each urination were recorded using the automated Voided Stain On Paper (aVSOP) method for four consecutive days (Fig. 1b and

Supplementary Fig. 1). Based on these recordings, urine volume was measured in each urine spot and the total amount of urine in each 4 h period was used to show urination behavior. Since mice are nocturnal, we calculated the diurnal polyuria index in mice (Diurnal Polyuria index: DPi refers to the ratio of diurnal urine volume to daily urine volume), which was used as a corresponding index for nocturnal polyuria in humans. As expected, salt loading did not change DPi in young mice (Fig. 1c, 0.12 vs. 0.13, n.s.). Further, no changes in inactive period urine volume and daily urine volume were observed, altogether indicating that salt loading did not lead to nocturnal polyuria in young mice. In contrast, salt loading increased Dpi in aged mice (Fig. 1d 0.22 vs. 0.30, $p < 0.05$). An increase in inactive period urine volume was also observed with no change in daily urine volume. Taken together, only aged mice exhibited nocturnal polyuria after salt loading, as observed in humans. Histological evaluation of the kidney revealed glomerulosclerosis, tubular injury, and interstitial fibrosis exclusively in aged mice (Fig. 1e and Supplementary Fig. 2). In order to investigate the age-related factors contributing to nocturnal polyuria in old mice, we focused on nitric oxide (NO), as NO production decreases with age and is associated with age-related diseases including hypertension[22], coronary artery disease[23], and cerebral infarction[24]. In order to assess the association between nocturnal polyuria and NO production in humans, we collected 24-h urine and measured urinary $NO_2/NO_3$ ($=NOx$) levels, which reflect the amount of NO production (Supplementary Table 1). No association was found between dietary salt intake and nocturnal polyuria (Supplementary Fig. 3). However, when the subjects were divided into high and low NOx level groups, a strong correlation was observed between salt intake and the nocturnal polyuria index (the ratio of nocturnal urine volume to daily urine volume) in subjects with low urinary NOx levels ($r = 0.645$, $p < 0.05$) (Fig. 1f, left). On the other hand, no correlation was observed in those with high urinary NOx ($r = -0.097$, n.s.) (Fig. 1f, right). A comparison of the correlation coefficients revealed a significant difference ($p < 0.05$). To compare NO production in mice, we measured NOx levels in 24-h urine. Urinary NOx in aged mice was significantly lower than in young mice (Fig. 1g). These results suggest that reduced NO production affects the relationship between salt intake and nocturnal polyuria in humans and mice.

### A 'two-hit' mouse model of nocturnal polyuria.
Based on the above results, we formulated a "two-hit hypothesis", wherein concomitant low NO levels and high salt intake induce nocturnal polyuria. To test this hypothesis, we divided young mice (19 weeks old) into four groups receiving tap water or L-NAME ($N^\omega$-Nitro-L-arginine methyl ester hydrochloride: NO synthase inhibitor, 5 mg/dL in drinking water) in parallel to the NSD or HSD (Fig. 2a). Urinary NOx was significantly decreased by L-NAME administration (Fig. 2b). There was no significant difference in daily urine volume between groups (Fig. 2c, d). Salt loading and tap water did not change the DPi (0.12 vs. 0.13, n.s.) nor the inactive period urine volume, while salt loading and L-NAME administration increased DPi (0.23 vs. 0.28, $p < 0.05$) and inactive period urine volume (439 vs. 655 μl, $p < 0.05$). Significant differences in the increase of DPi after high salt loading were observed between the groups treated with and without L-NAME (two-way ANOVA, $p < 0.05$) (Supplementary Fig. 4). These results indicated that L-NAME + HSD induces nocturnal polyuria. To examine whether L-NAME + HSD resulted in other nocturnal polyuria manifestations, such as excessive food and water intake, impaired renal function, insomnia, and hypertension, we assessed body weight, daily food and water intake, serum Cr and Na, blood pressure, renal histology, and behavioral

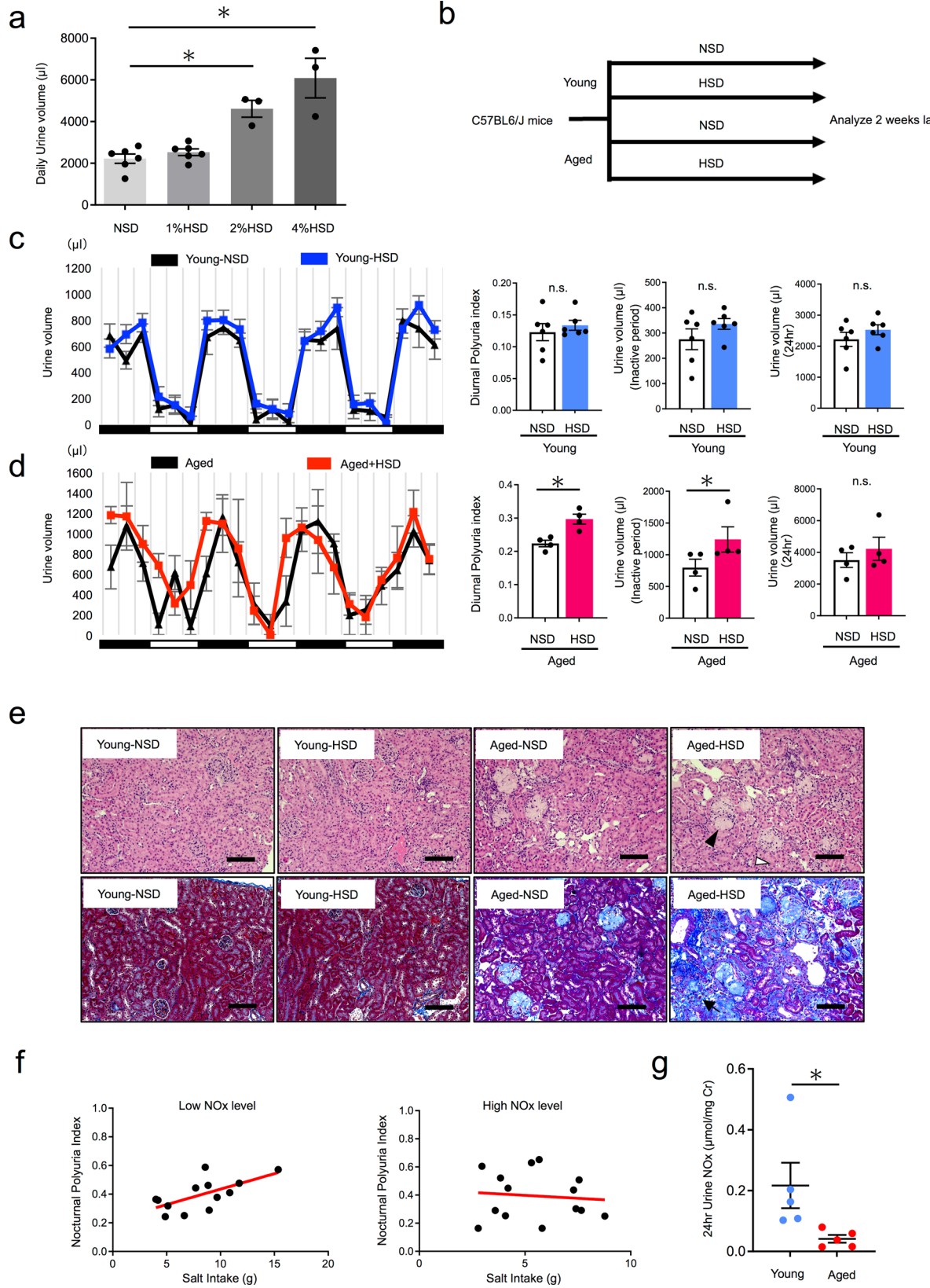

patterns during the inactive period. There were no significant differences in body weight, food intake, water intake, serum Na concentration (Table 1), or behavior patterns during the inactive period among the four groups (Supplementary Fig. 5). L-NAME treatment resulted in a significant decrease (23%) in serum Cr levels (Table 1), consistent with renal function decline in elderly

subjects[25]. Histopathological analysis of the kidney revealed no glomerulosclerosis, tubular atrophy, or fibrosis of the interstitium in all four groups (Fig. 2e). Systolic blood pressure was significantly elevated after L-NAME compared to tap water (Table 1). To investigate whether elevated blood pressure causes nocturnal polyuria seen under L-NAME + HSD, we assessed

**Fig. 1 The effect of salt loading on diurnal polyuria differs in young and aged mice. a** Daily urine volume by salt concentration (NSD, 1% HSD, 2% HSD, 4% HSD). NSD: 0.2% Salt diet, 1% HSD: 1% Salt diet, 2% HSD: 2% Salt diet, 4% HSD: 4% Salt diet (NSD,1% HSD: $n = 6$ mice per group, 2% HSD, 4% HSD: $n = 3$ mice per group). Statistical analysis was performed using the Tukey–Kramer method **b** Experimental protocol. **c** Mean 4-h urinary volume for three consecutive days in young NSD (black) and young HSD mice (blue) (19 weeks old), as measured via automated voided stain on paper (aVSOP). Daily urine volume, diurnal urine volume, and DPi (diurnal polyuria index = diurnal urine volume/daily urine volume). The active period (night period) was from 8:00 p.m. to 8:00 a.m. and the inactive period (daytime period) was from 8:00 a.m. to 8:00 p.m. (both 12 h). DPi was calculated by averaging over three days ($n = 6$ mice per group). Statistical analysis was performed using the two-tailed Student's $t$-test. **d.** Mean 4-h urinary volume for three consecutive days in aged NSD (black) and HSD mice (red) (80 weeks old). Daily urine volume, diurnal urine volume, and DPi in aged NSD and HSD mice (80 weeks old) ($n = 4$ mice per group). Statistical analysis was performed using the two-tailed Student's $t$-test. **e.** Representative images of hematoxylin and eosin histological staining (above) and Masson Trichrome staining (below) of kidney tissue from a young and aged mice (young NSD, young HSD, aged NSD, aged HSD). Glomerulosclerosis (black arrowhead), tubular atrophy (white arrowhead), and interstitial fibrosis (black arrow). Original magnification: ×200. Scale bar, 100 μm. **f** Correlation between dietary salt intake and nocturnal polyuria index in subjects with low and high urinary NOx levels. (NOx-low: $n = 13$, NOx-high: $n = 14$). Statistical analysis was performed using the Fisher $r$-to-$z$ transformation. **g** Urinary NOx in young and aged mice ($n = 5$ mice per group). Statistical analysis was performed using the two-tailed Student's $t$-test. All data are expressed as the mean ± SEM, *$p < 0.05$.

whether antihypertensive drugs would improve nocturnal polyuria. We administered an antihypertensive Ca channel blocker, amlodipine (6.7 mg/kg/day in drinking water), to L-NAME + HSD group mice. Amlodipine significantly decreased systolic blood pressure (Fig. 2f, left, 122 vs. 85 mmHg, $p < 0.05$). While amlodipine did not change daily urine volume (Fig. 2g, right), it increased the DPi and inactive period urine volume (Fig. 2g left, middle 0.281 vs. 0.428, $p < 0.05$), indicating that antihypertensive drugs do not improve nocturnal polyuria. These results suggest that elevated blood pressure does not cause nocturnal polyuria. Taken together, we developed a 'two-hit' mouse model of nocturnal polyuria that recapitulates clinical features.

**Overactivation of NCC during the active period inhibits sodium excretion.** Osmotic substances such as urea nitrogen (UN), sodium (Na), and potassium (K) play a critical role in urine volume regulation[26]. We previously found that urinary Na excretion was the most relevant of these in a clinical study of nocturnal polyuria[18]. Therefore, we compared urinary Na excretion among the four groups. Salt loading increased 24-h Na excretion to the same extent in groups with and without L-NAME (Fig. 3a, left). Tap water + HSD significantly increased Na excretion by +0.08 mEq during the active period (Fig. 3a and Supplementary Table 2). Excretion remained almost unchanged (+0.01 mEq) during the inactive period (Fig. 3a and Supplementary Table 2). L-NAME + HSD induced a significant increase in Na excretion during the active (+0.06 mEq) and the inactive period (+0.04 mEq) (Fig. 3a and Supplementary Table 2). A significant difference in the increase of salt excretion was observed between the groups with and without L-NAME during both the active and inactive period (two-way ANOVA, $p < 0.05$) (Fig. 3a). These results suggest that salt loading under L-NAME attenuates the increase in urinary sodium excretion during the active period and enhances the increase in excretion during the inactive period.

To investigate the molecular mechanisms underlying changes in urinary sodium excretion, we evaluated the expression of sodium chloride co-transporter (NCC) and epithelial sodium channel (ENaC) in the active phase, which are the main regulators of the sodium balance, in the distal tubule and collecting duct, respectively[27]. The expression of total NCC was not altered after salt loading or salt loading + L-NAME (Fig. 3b, c). The expression of phosphorylated (active) NCC decreased in response to salt loading (0.99 vs. 0.68, $p < 0.05$), thus stimulating salt excretion, while salt loading + L-NAME did not decrease the expression of phosphorylated NCC (0.86 vs. 0.78, n.s.). A significant difference was observed in phosphorylated NCC levels in response to salt loading, with or without L-NAME. ENaCα, ENaCβ and ENaCγ expression was not altered after salt loading,

regardless of L-NAME. Taken together, these results indicate that reduced NO production leads to overactivation of NCC even under salt loading, decreasing urinary Na excretion during the active period. Subsequently, Na excretion increases during the inactive period to compensate, leading to nocturnal osmotic diuresis and nocturnal polyuria.

**Nocturnal polyuria was improved via NCC inhibition but not ENaC inhibition.** These findings led us to hypothesize that nocturnal polyuria is mediated by NCC overactivation. To clarify whether nocturnal polyuria is improved by suppressing NCC, we administered hydrochlorothiazide (HCTZ 20 mg/kg s.c.), which inhibits NCC activity, to the NSD and L-NAME + HSD groups and compared urine volume and Na excretion between active and inactive periods. Neither the active nor inactive period urine volume of the NSD group was altered by HCTZ administration (Fig. 4a). In contrast, the L-NAME + HSD group exhibited an increase in active period urine volume and a decrease in inactive period urine volume, thus resulting in a decrease in DPi (0.28 vs. 0.19, $p < 0.05$). Further, HCTZ treatment in the L-NAME + HSD group increased sodium excretion during the active period, while no significant change was observed during the inactive period (Fig. 4b and Supplementary Table 3). Thus, inhibition of NCC via HCTZ increased active period sodium excretion in the nocturnal polyuria model, accompanied by an increase in active period urine volume, which in turn decreased inactive period volume and DPi. These results indicate that nocturnal polyuria is mediated through NCC hyperactivation, and blocking NCC represents a therapeutic strategy for nocturnal polyuria. To investigate whether nocturnal polyuria is mediated via ENaC, we administered amiloride (1.45 mg/kg/day subcutaneously), which inhibits ENaC activity, to mice and compared urine volume. In both NSD and L-NAME + HSD groups, amiloride did not change the inactive period urine volume, 24-h urine volume, and DPi (0.12 vs. 0.13 vs. 0.28 vs. 0.25, n.s.) (Fig. 4c). This indicates that ENaC activity is not implicated in the nocturnal polyuria mouse model.

**Angiotensin II in the kidney activates the SPAK–NCC pathway.** To further elucidate the molecular mechanisms underlying nocturnal polyuria and to explore new therapeutic targets, we examined signaling upstream of NCC. We first determined the protein expression of SPAK (STE20/SPS1-related proline–alanine-rich protein kinase), which phosphorylates and activates NCC[28], as well as that of its active form, phosphorylated SPAK. SPAK expression was not altered after salt loading or salt loading under L-NAME (Fig. 5a). Phosphorylated SPAK significantly decreased after salt loading under tap water intake, which was not observed under L-NAME. These results suggest

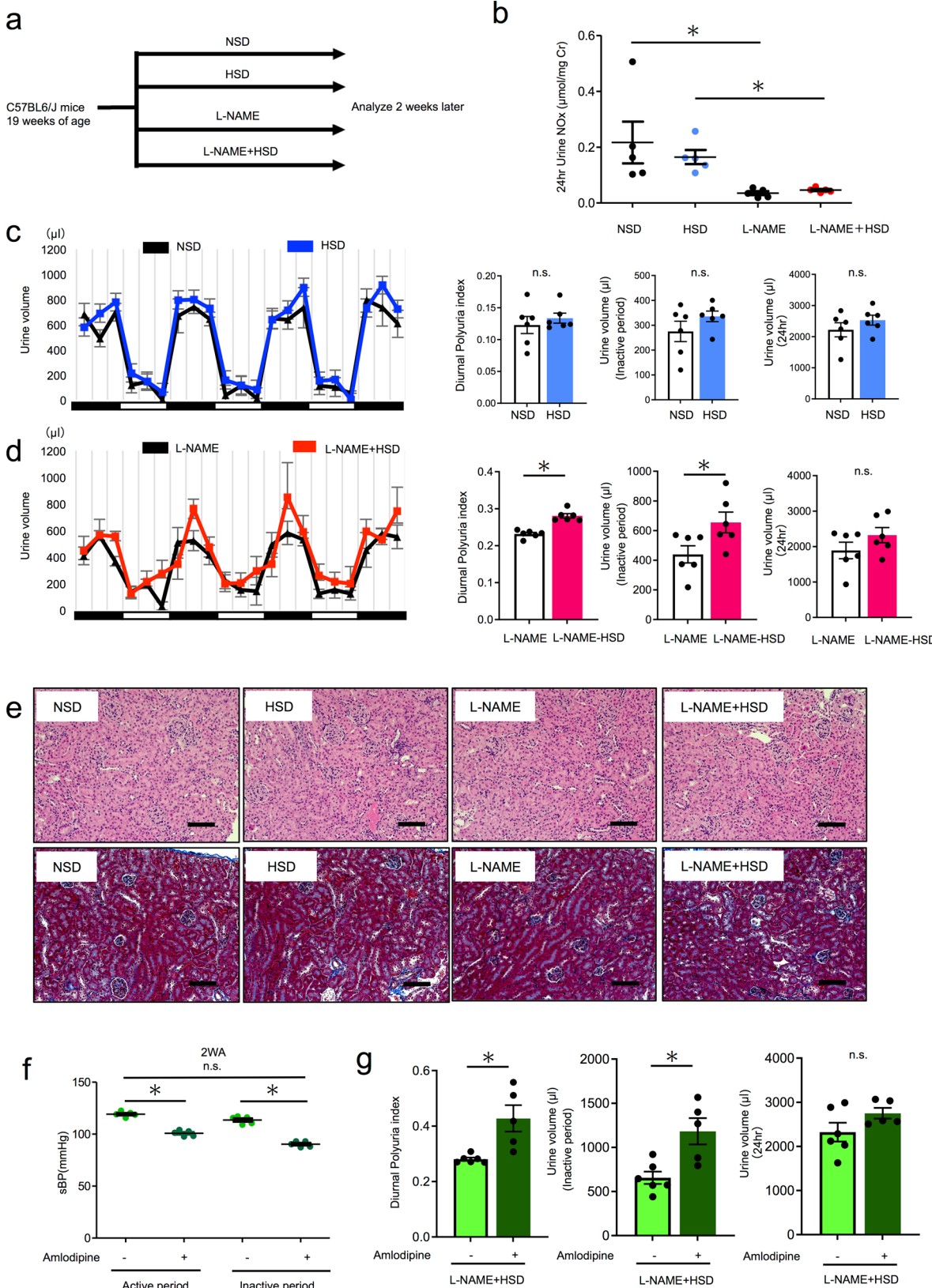

that reduced NO production hyperactivated SPAK and, consequently, NCC (the SPAK–NCC pathway) during the active period. Pathway activation was not diminished by salt loading, leading to nocturnal polyuria.

SPAK–NCC pathway activation has been reported to occur through a variety of mechanisms, including aldosterone,

angiotensin II, and insulin signaling as well as extracellular K and oxidative stress[29–33]. We examined whether aldosterone and angiotensin II, which are generally implicated in the effects of salt loading, are indeed involved. Recently, the intrarenal local renin–angiotensin system (RAS) system, independent of the systemic RAS system, has been suggested to play a role in the

**Fig. 2 A 'two-hit' mouse model of nocturnal polyuria. a** Experimental protocol. **b** Urinary NO$x$ concentration adjusted by urinary creatinine level in NSD, HSD, L-NAME, and L-NAME + HSD groups ($n = 5$). Statistical analysis was performed using the Tukey–Kramer method. **c** Mean 24-h urinary volume for three consecutive days in young NSD (black) and young HSD mice (blue) (19 weeks old), as measured via aVSOP. Daily urine volume, diurnal urine volume, and DPi (Diurnal polyuria index = diurnal urine volume/daily urine volume). The active period (night period) is from 8:00 p.m. to 8:00 a.m. and the inactive period (daytime period) is from 8:00 a.m. to 8:00 p.m. (both are 12 h). DPi was calculated by averaging over 3 days ($n = 6$ mice per group). Statistical analysis was performed using the two-tailed Student's $t$-test. **d** Mean 24-h urinary volume for three consecutive days in L-NAME (black) and L-NAME + HSD mice (red) (19 weeks old). Daily urine volume, diurnal urine volume, and DPi under L-NAME administration (5 mg/dL in drinking water) ($n = 6$ mice per group). Statistical analysis was performed using the two-tailed Student's $t$-test. **e** Representative images of H&E histological staining (above) and Masson Trichrome staining (below) of kidney tissue from young mice (NSD, HSD, L-NAME + NSD, L-NAME + HSD). Glomerulosclerosis (black arrowhead), tubular atrophy (white arrowhead), and interstitial fibrosis (black arrow). Original magnification: ×200. Scale bar, 100 μm **f** Effect of amlodipine (Ca channel blocker) on sBP (systolic blood pressure). sBP was evaluated in active (10 p.m.) and inactive period (10 a.m.) by tail-cuff method (sBP: $n = 5$ mice per group). **g** Effect of amlodipine (Ca channel blocker) on diurnal polyuria index (DPi), inactive period urine volume, and 24-h urine volume. (NSD group: $n = 6$, L-NAME + HSD group: $n = 5$) All data are expressed as the mean ± SEM, *$p < 0.05$.

**Table 1 Physiological data of the four groups.**

|  | NSD | HSD | L-NAME | L-NAME + HSD |
|---|---|---|---|---|
| Body weight (g) | 29.6 ± 0.68 | 29.8 ± 0.67 | 29.2 ± 0.33 | 29.1 ± 0.82 |
| *Food intake (g)* |  |  |  |  |
| Active phase | 3.58 ± 0.32 | 3.22 ± 0.26 | 3.32 ± 0.40 | 3.50 ± 0.40 |
| Inactive phase | 0.52 ± 0.19 | 0.66 ± 0.29 | 0.58 ± 0.18 | 0.74 ± 0.19 |
| Total | 4.10 ± 0.45 | 3.88 ± 0.37 | 3.90 ± 0.27 | 4.24 ± 0.27 |
| *Water intake (g)* |  |  |  |  |
| Active phase | 4.42 ± 0.22 | 4.30 ± 0.42 | 4.34 ± 0.49 | 4.66 ± 0.39 |
| Inactive phase | 0.60 ± 0.12 | 0.72 ± 0.19 | 0.76 ± 0.18 | 0.66 ± 0.09 |
| Total | 5.02 ± 0.28 | 5.02 ± 0.28 | 5.10 ± 0.48 | 5.32 ± 0.36 |
| *Systolic blood pressure (mmHg)* |  |  |  |  |
| Active phase | 107 ± 1.33 | 110 ± 2.76 | 112 ± 4.67 | 119 ± 2.50* |
| Inactive phase | 96 ± 2.69 | 99 ± 4.19 | 103 ± 4.28 | 114 ± 3.60* |
| Serum Na (mEq/L) | 149 ± 2.99 | 148 ± 0.93 | 150 ± 0.98 | 152 ± 1.85 |
| Serum Cr (mg/dL) | 0.13 ± 0.01 | 0.13 ± 0.01 | 0.16 ± 0.01* | 0.17 ± 0.02* |

Body weight, food intake, water intake, systolic blood pressure, as well as serum Na and Cr levels. Data are expressed as the mean ± SEM. Statistical analysis was performed using the Tukey–Kramer method. Systolic blood pressure: L-NAME + HSD vs. NSD or HSD or L-NAME, Serum Cr: L-NAME + HSD vs. NSD, L-NAME vs. NSD.
*$p < 0.05$ ($n = 5$ mice per group).

development of hypertension[34–36] and CKD[37,38]. All molecules necessary for the production of biologically active angiotensin II, such as angiotensinogen, renin, and angiotensin-converting enzyme, are present in the kidney. Renal angiotensinogen, a substrate for the production of angiotensin II, is utilized for the local production of angiotensin II in the kidney[39–43]. Thus, we compared serum aldosterone levels and the protein levels of renal angiotensinogen between the four groups. Serum aldosterone levels decreased after salt loading both under tap water and L-NAME administration (Fig. 5b, 34.1 vs. 13.3 and 103.0 vs. 29.1, respectively). Renal angiotensinogen did not change after salt loading under tap water (Fig. 5c, 1.01 vs. 1.09), whereas there was a marked increase in renal angiotensinogen after salt loading under L-NAME (Fig. 5c, 1.44 vs. 2.77). These results indicate that salt loading decreased serum aldosterone under physiological conditions, which suppressed the SPAK–NCC pathway and led to the urinary excretion of sodium. In contrast, salt loading under reduced NO production decreased serum aldosterone, but not renal angiotensin II, and subsequently hyperactivated the SPAK–NCC pathway, leading to an insufficient urinary excretion of sodium and nocturnal polyuria (Fig. 5d).

## Discussion

Nocturnal polyuria is prevalent in middle-aged and elderly people and is the most common cause of nocturia, which impairs the quality of life of patients and is associated with increased mortality[2,44]. Its underlying mechanism is not well elucidated, partly owing to the limitation of current preclinical models. In this study, we developed a novel mouse model of nocturnal polyuria that recapitulates the clinical features of nocturnal polyuria by combining two intervention methods—high-salt diet and NO suppression—based on previous pathophysiological observations in humans. Using this mouse model, we were able to uncover one of the molecular mechanism underlying nocturnal polyuria. In summary, excessive salt intake along with NCC overactivation by NO suppression promotes insufficient sodium excretion in the active period, whereas increased sodium excretion during the inactive period causes osmotic diuresis, thereby leading to nocturnal polyuria.

To date, epidemiological, observational, and interventional clinical studies in human patients have been conducted in an attempt to elucidate the condition's underlying mechanisms[3,44–46]. In general, nocturnal polyuria occurs as a result of water retention in the body during the day, such as following excessive water intake[47,48], or accumulation of interstitial fluid in the subcutaneous tissue, which is excreted as urine at night[49]. Another proposed mechanism is the lower urinary concentration during the night resulting from decreased renal function[50] or impaired responsiveness of vasopressin[51], which plays an integral role in renal water reabsorption, thus increasing nocturnal urine output[51]. Human studies have revealed that osmotic substances such as UN, Na, and K are major contributing factors to nocturnal polyuria, as osmolyte excretion is responsible for half of nocturnal polyuria cases[16,26], and excess salt intake proportionally increases the risk of nocturnal polyuria[16]. To elucidate the role of osmolyte excretion in nocturnal polyuria, we previously examined urinary osmolyte excretion before and after surgery in nephrectomized patients and revealed that postoperative nocturnal polyuria was associated with

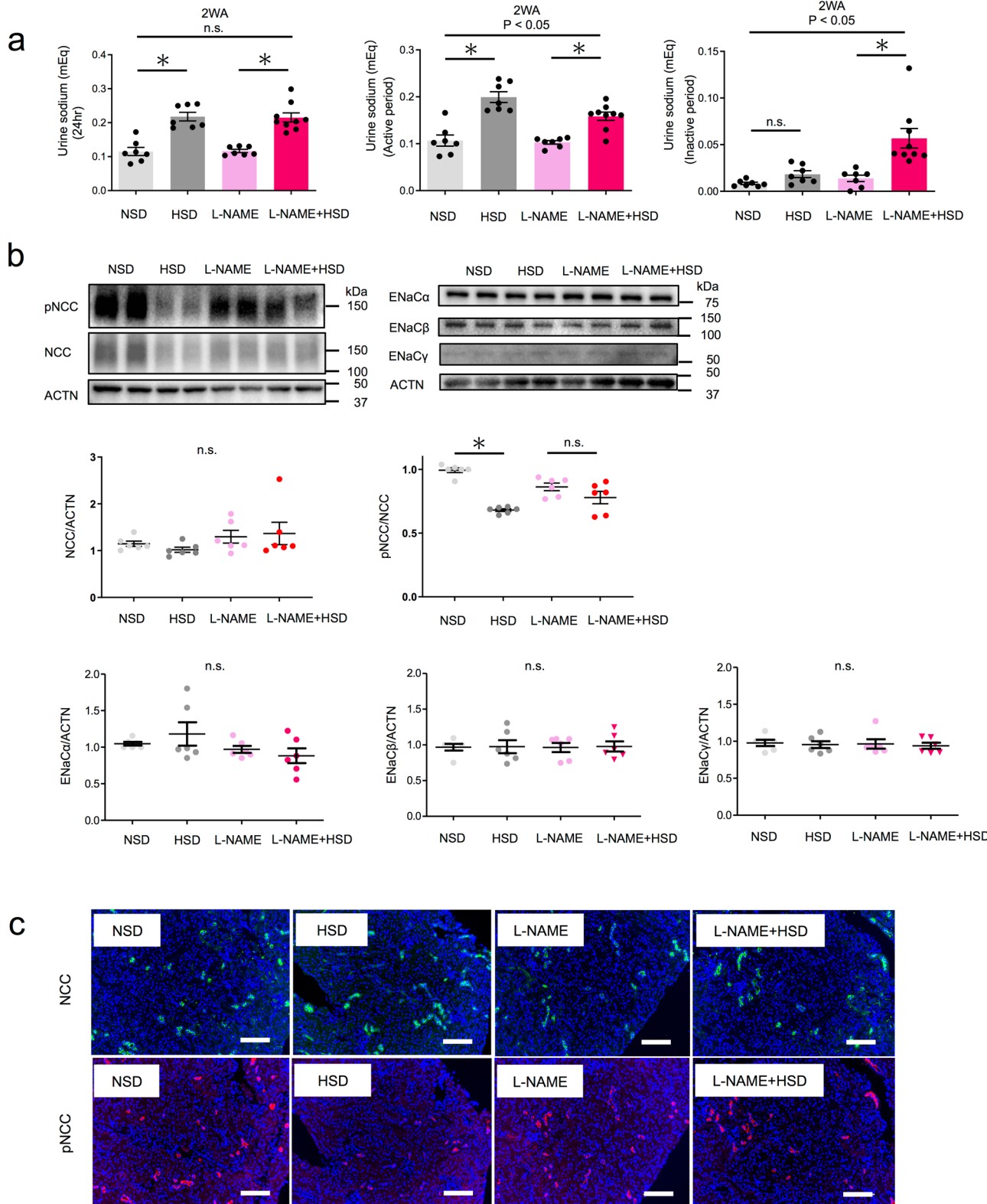

**Fig. 3 Overactivation of NCC during the active period inhibits sodium excretion under NO suppression. a** Urinary sodium excretion per day (left) during active (middle) and inactive (right) periods (NSD, HSD, L-NAME: $n = 7$, L-NAME + HSD: $n = 9$). Statistical analysis was performed using the two-way ANOVA or the two-tailed Student's $t$-test. **b** Representative immunoblotting (above) and quantitative analysis (below) of renal phosphorylated NCC (pT53), NCC, ENaCα, ENaCβ, ENaCγ, and ACTN in the active phase for the four groups. Data are expressed as the mean ± SEM. *$p < 0.05$ ($n = 6$ mice per group). Statistical analysis was performed using two-way ANOVA. **c** Representative fluorescent immunostaining of renal phosphorylated NCC (pT53) and total NCC for the four groups using serial sections. Original magnification: ×200. Scale bar, 100 μm. All data are expressed as the mean ± SEM, *$p < 0.05$.

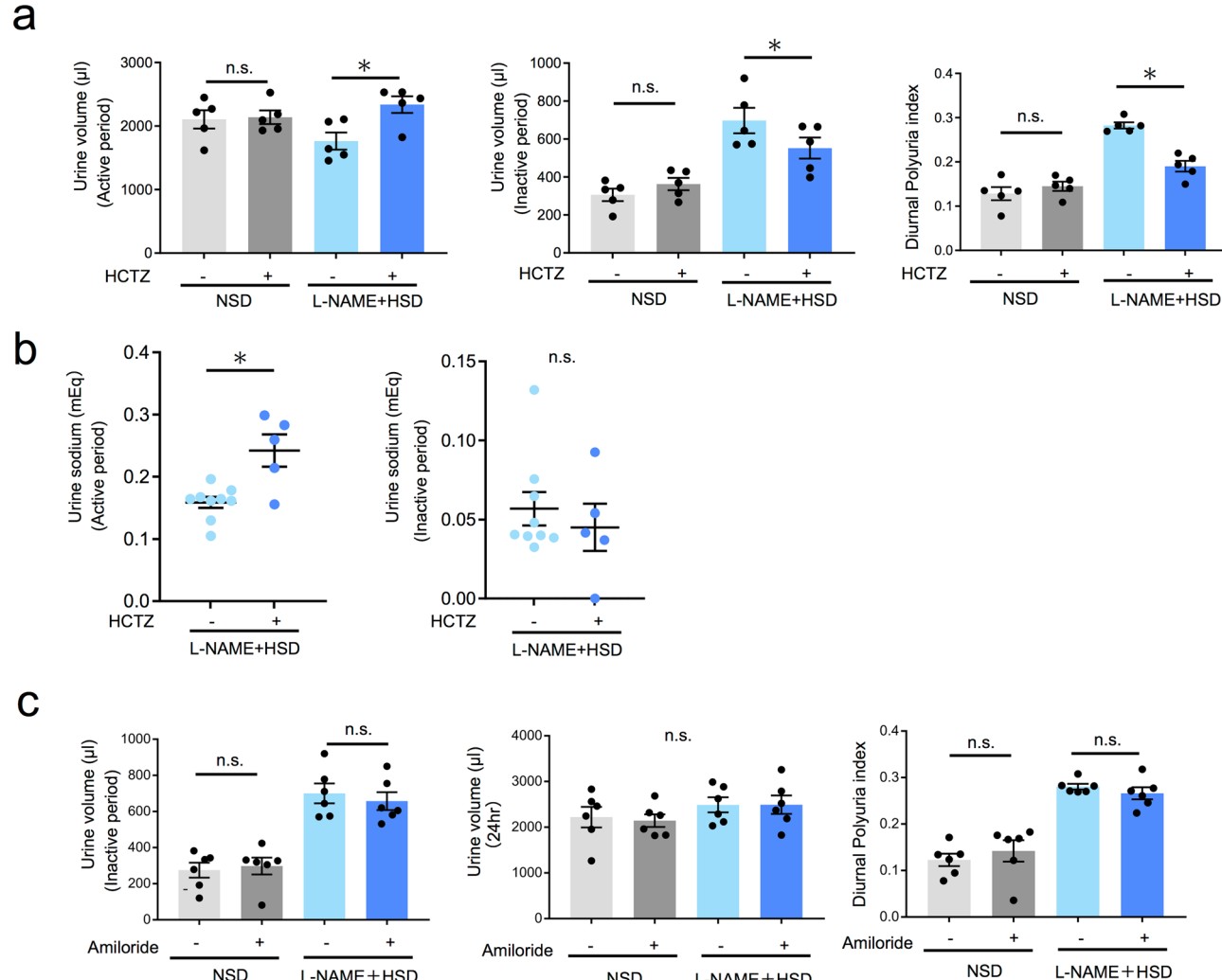

**Fig. 4 Nocturnal polyuria is improved by treatment with an NCC inhibitor (HCTZ), but not by ENaC inhibitor treatment (amiloride). a** Active period urine volume and inactive period urine volume and DPi in NSD and L-NAME + HSD groups after HCTZ administration ($n = 5$ mice per group). Statistical analysis was performed using the two-tailed Student's $t$-test. **b** Urinary sodium excretion per day during active (left) and inactive (right) periods (HCTZ-: $n = 9$, HCTZ+: $n = 5$ mice per group). Statistical analysis was performed using two-way ANOVA or the two-tailed Student's $t$-test. *$p < 0.05$. **c** Inactive period urine volume and 24-h urine volume and DPi in NSD and L-NAME + HSD groups after amiloride administration. Data are expressed as the mean ± SEM (amiloride-: $n = 6$, amiloride+: $n = 5$). Statistical analysis was performed using the two-tailed Student's $t$-test. All data are expressed as the mean ± SEM, *$p < 0.05$.

nocturnal Na excretion but not with nocturnal UN or decreased urine concentration[18]. These previous findings indicate that a shift in Na excretion shift from day to night contributes to nocturnal polyuria. However, the mechanism through which Na excretion shifts has not been well understood. In the present study, we found that NCC activation in the distal tubule plays a crucial role in nocturnal polyuria. Excessive salt intake under physiological conditions decreased NCC activity, resulting in increased urinary sodium excretion during the active period. In contrast, when NCC was hyperactivated during the active period, the excretion of surplus sodium was carried over to the inactive period, leading to nocturnal polyuria. Under normal salt intake, sodium is excreted during the active phase even during NCC activation, and nocturnal polyuria does not occur.

In addition, our study revealed that the overactivation of NCC is triggered by reduced NO production. In humans, nocturnal polyuria occurs mainly in the elderly[21], even though salt intake is not significantly different between the young and the elderly. The findings of our mouse study were in accordance with this phenomenon. In a previous report[47], nocturnal polyuria occurred

more frequently in the elderly than in younger adults under excessive water intake. These observations suggest that age is a significant determinant of physiological water and salt dynamics. In the present study, we focused on NO, as age-associated NO suppression has been implicated in a variety of age-related conditions[22,51]. NO is produced by NO synthase in vascular endothelial cells and nerves, exerting its effects on the surrounding structures through diffusion[52]. In the kidney, NO has numerous functions, including the regulation of renal hemodynamics, maintenance of medullary perfusion, mediation of pressure-natriuresis, blunting of tubulo-glomerular feedback, and modulation of renal sympathetic neural activity[52]. We found that excessive salt intake and decreased NO production in human subjects promoted nocturnal polyuria, suggesting that NO is involved in the regulation of salt and water excretion. Based on these results, we developed a mouse model in which simultaneous low NO and excessive salt intake also resulted in nocturnal polyuria. Furthermore, we confirmed that suppressed NO production leads to NCC overactivation. In the absence of NO suppression, salt loading lowered aldosterone levels and NCC

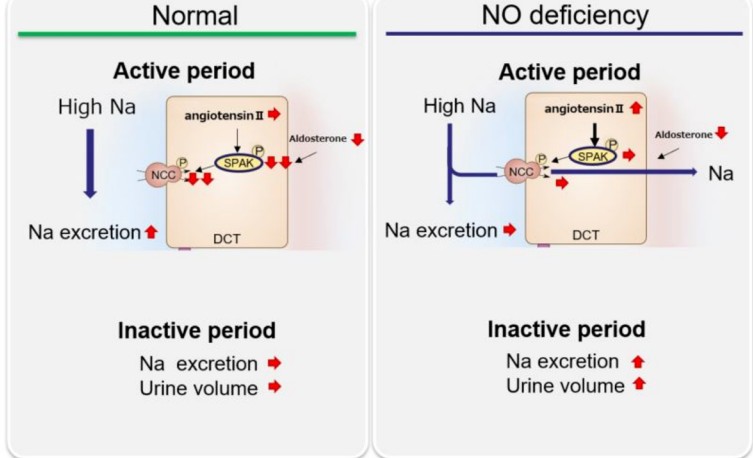

activity, resulting in increased salt excretion. However, when NO production was suppressed, local angiotensin activated the SPAK-NCC pathway despite the decrease in aldosterone levels, in turn resulting in lower salt excretion. These results suggested that NO plays a pivotal role in the relationship between salt intake and nocturnal polyuria.

The kidneys play an essential role in the precise regulation of blood composition by controlling the glomerular filtration rate (GFR) and differentially transferring electrolytes as well as water into the urine through various channels[53]. In addition to NCC in the distal tubule, sodium regulation involves Na+/H+ exchanger 3 (NHE3) in the proximal tubule, Na–K–2Cl co-transporter 2

**Fig. 5 Intrarenal angiotensin II activates the SPAK–NCC pathway. a** Representative immunoblotting (above) and quantitative analysis (below) of renal phosphorylated SPAK and total SPAK in the active phase in the four groups ($n = 6$ mice per group). Statistical analysis was performed using two-way ANOVA. **b** Serum aldosterone levels among the four groups (NSD, HSD, L-NAME + HSD: $n = 5$, L-NAME: $n = 4$). Statistical analysis was performed using the Tukey–Kramer method. **c** Representative immunoblotting (above) and quantitative analysis (below) of renal AGT (angiotensinogen) in the active phase among the four groups. Each point is presented as the mean ± SEM. *$p < 0.05$ ($n = 6$ mice per group). Statistical analysis was performed using two-way ANOVA. **d** Schematic summary illustrating the regulation of sodium and urine via the angiotensin II-SPAK–NCC pathway in the kidney of our novel nocturnal polyuria mouse mode. All data are expressed as the mean ± SEM, *$p < 0.05$.

(NKCC2) in Henle's loop, and ENaC in the collecting duct, while water regulation involves aquaporin (AQP) in the collecting duct[54]. The water and electrolyte balance is controlled by the GFR, which is modulated through a tubulo-glomerular feedback mechanism wherein macra densa cells at the origin of the distal tubule sense chloride ions, and afferent arterioles respond[55]. Under low or normal salt loading, ~99% of the sodium filtered through the glomerulus is reabsorbed in the urinary tubule, 90% of it being reabsorbed in the proximal tubule and Henle's loop and only about 10% reabsorbed in the distal tubule or further down the renal route[56]. During excessive salt loading, inhibition of reabsorption by Na channels in the distal tubule as well as through NCC and ENaC rather than Na channels in the proximal tubule and Henle's loop, plays a role. In this study, we investigated whether NCC and ENaC function as key regulators of Na excretion and osmotic diuresis in response to excess salt. We found that NCC, but not ENaC, plays an important role in the regulation of urinary sodium excretion and subsequent osmotic diuresis. Consistent with our observations, NCC activation drives salt-sensitive hypertension in chronic kidney disease[29], while other groups suggest that ENaC activity is key to the development of hypertension in spontaneously hypertensive rats[57], indicative of disease-specific involvement.

Excessive salt intake does not always increase urine volume because of a mechanism that suppresses urine volume by controlling osmolytes. For example, when increasing Na excretion, UN, and K excretion are reduced at the same time, keeping the total osmolyte content constant[58]. In our study, tap water + HSD increased Na excretion (+0.08 mEq, 72% increase) (Supplementary Table 2) during the active period, but the urine volume did not increase as much as Na increased through concentrating urine (+249 mL, 13% increase). On the other hand, salt loading under L-NAME administration resulted in a two-fold lower yet still significant increase (+0.04 mEq, 200% increase) in Na excretion during the inactive phase, in addition to a simultaneous increase in urine volume (+216 mL, 49% increase) as the amount of Na that could be eliminated with the initial urine volume might have been exceeded. A higher urine volume results in a correspondingly higher amount of Na that can be excreted, while a lower urine volume decreases the amount of Na that can be excreted. That is, due to the low inactive urine volume, even a small increase in salt excretion commands an increase in urine volume. This was illustrated by the fact that after tap water + HSD, the urinary Na concentration increased to 87 mEq/L during the active phase (Supplementary Table 2), producing concentrated urine, while the urinary Na concentration during the inactive phase after L-NAME + HSD also increased to 92 mEq/L, yielding concentrated urine.

We found that the angiotensin II–SPAK–NCC pathway is activated in our nocturnal polyuria mouse model, potentially representing a new therapeutic target. NCC inhibitor thiazide improved nocturnal polyuria by stimulating sodium excretion during the day. In the clinical setting, nocturnal polyuria was improved by the additional administration of thiazide as a diuretic in patients with nocturia who did not respond to alpha-1 blockers[59]. While thiazide was used as a diurnal urine volume enhancer in the above-mentioned clinical study, our findings suggests that it may also play a role in diurnal salt excretion. Identification of patients with nocturnal polyuria who also have low NO levels could allow for more effective treatment through thiazide. In addition to thiazide, RAS system inhibitors, such as angiotensin II receptor blockers and antioxidants, which suppress the angiotensin II–SPAK–NCC pathway, may be candidates for treatment.

The current study had several limitations. First, we did not measure the glomerular filtration rate (GFR), so we could not evaluate the effects of L-NAME administration or salt loading on GFR. GFR is an important determinant of urine output. Both L-NAME and salt loading can alter GFR and potentially lead to polyuria. Second, we did not evaluate other Na transporters such as NHE3 and NKCC2 nor water channels such as aquaporin. Further studies are required to address these. Third, we used the tail-cuff method to evaluate blood pressure, but measurement of blood pressure for 24 h using telemetry would be better to assess the relationship between blood pressure and nocturnal polyuria in detail. Fourth, the volume of urine during the inactive period was small, so urine volume and urinary electrolyte concentration were calculated.

In summary, we developed a mouse model of nocturnal polyuria that recapitulates the clinical features observed in patients. We then utilized this mouse model to explore a previously unknown regulatory mechanism of urine production rhythm in the middle-aged and elderly. We clarified that high salt intake under conditions of reduced NO production hyperactivates the angiotensin II–SPAK–NCC pathway in the kidney, resulting in the insufficient excretion of sodium during the day and nocturnal polyuria. Thus, the intrarenal angiotensin II–SPAK–NCC pathway may represent a promising therapeutic target for nocturnal polyuria.

## Methods

**Experimental animals.** All experiments involving animals were conducted in accordance with the Guidelines for the Care and Use of Laboratory Animals of Osaka University and were approved by the Animal Care and Use Committee of Osaka University (No. J006580-013). The studies complied with all ethical regulations. Nineteen-week-old C57BL/6J mice (young mice) were purchased from Japan SLC, and 80-week-old C57BL/6J mice (aged mice) were purchased from Charles River Laboratories Japan. They were kept in a temperature-controlled room with 12/12-h light/dark cycle and had ad libitum access to food and water. Young mice were randomly assigned to a normal-salt diet (NSD; 0.2% wt/wt NaCl) or a 1% high-salt diet (1%HSD; 1% wt/wt NaCl), 2% high-salt diet (2% HSD; 2% wt/wt NaCl), and 4% high-salt diet (4% HSD; 4% wt/wt NaCl). All diets were purchased from Oriental Yeast (Tokyo, Japan). After 2 weeks, we measured urine volume via the aVSOP method. Young mice and aged mice were randomly assigned to NSD or HSD (1% high-salt diet) with tap water or L-NAME (0.5 g/L in drinking water) for 2 weeks. L-NAME was purchased from Sigma-Aldrich. After 2 weeks, the urine volume was measured once again. Mice were then sacrificed in the active phase by inhalation of isoflurane, and their kidneys as well as blood and urine samples were collected.

**Drug administration.** L-NAME (5 mg/dL, N5751, Sigma-Aldrich, St. Louis, MO, USA) and amlodipine (6.7 mg/kg/day, A2353, Tokyo Chemical Industry, Japan) were dissolved in drinking water. Hydrochlorothiazide (20 mg/kg/day, H4759, Sigma-Aldrich) and amiloride (1.45 mg/kg/day, A2599, Tokyo Chemical Industry, Japan) was administered via subcutaneous injection.

**Urine volume and time measurements**. Measurements were performed via the automated Voided Stain On Paper (aVSOP) method[60,61]. Briefly, a roll of laminated filter paper, pretreated to turn urine dark purple, was rolled up under a water-repellent wire grid at a speed of 10 cm/h. Mice were housed for 4 days in cages with dimensions of 110 mm × 160 mm × 75 mm ($H \times D \times W$). Urination was counted, tracked, and converted to volume using ImageJ software (ver. 1.53e, National Institutes of Health, USA). The diurnal polyuria index was calculated by dividing the volume of urine during the inactive period (8 a.m. to 8 p.m.) by the volume of urine per day.

**Mouse urine analysis**. Mice were individually placed in metabolic cages (3600M021, Techniplast, Tokyo, Japan) with free access to food and water, and urine samples were collected. Since the volume of urine during the inactive period (daytime) was small, diurnal urine volume and diurnal urinary electrolyte concentration were calculated as follows: (inactive period urine volume) = (24-h urine volume) − (active period urine volume); (inactive period urinary electrolyte concentration) = {(24-h urinary electrolyte amount) − (active period urinary electrolyte amount)}/(inactive period urine volume). Urine electrolytes were measured using Fujifilm Pet Systems Co. Urinary NO$x$ (NO$_2$/NO$_3$) was measured using the NK05 NO$_2$/NO$_3$ Assay Kit-C II (Colorimetric, Japan) after separation of urine with a protein removal column and centrifugation at 7000×$g$ for 20 min, followed by centrifugation at 17,000×$g$ for 10 min.

**Human urine analysis**. Human samples were collected under protocols approved by the Institutional Review Board (No. 18418) at Osaka University, and consent to participate in this study was prospectively obtained in all cases. Patients characteristics were described in Supplementary Table 1. Urine was collected every 12 h from 10:00 to 22:00 and from 22:00 to 10:00, defined as daytime urine and nighttime urine, respectively. Urinary sodium (UNa) levels were measured during each period. The nocturnal polyuria index was defined as the ratio of nighttime urine volume to daily urine volume. Urinary NO$x$ (NO$_2$/NO$_3$) was measured as described above.

**Blood analysis**. Blood electrolyte and creatinine levels were measured by Fujifilm Pet Systems Co. Aldosterone levels were measured using the solid phase method by radioimmunoassay at the Japan Institute for the Control of Aging Co.

**Blood pressure measurement**. Blood pressure was measured using the tail-cuff method (BP-98A; Softron Corporation, Tokyo, Japan). Three consecutive measures were made, and the average value was taken.

**Behavioral patterns measurements**. The inactive period was divided into three equal parts (1–3). Behavioral patterns such as movement, food intake, and drinking were assessed in 1-h video recordings. Movement for more than 5 s was considered an action.

**Histological evaluation of kidney tissue**. Kidneys were fixed with 4% paraformaldehyde and embedded in paraffin. Histological examination was performed using HE staining or Masson trichrome staining. Glomerulosclerosis, tubular injury, and interstitial fibrosis were assessed based on scores from previous reports[62,63].

**Fluorescent immunostaining**. Fluorescent immunostaining was performed on 4-μm-thick serial sections of formalin-fixed, paraffin-embedded tissue. The slides were deparaffinized after heating at 68 °C for 20 min. Primary antibodies were added to each section and incubated overnight at 4 °C. The slides were then incubated with secondary antibodies for 60 min at room temperature. After a wash with TBS-T, the slides were counterstained with Prolong™ Gold Mountant with DAPI (Thermo Fisher Scientific). Immunofluorescence were examined using a keyence BZ-X700 microscope (Keyence, Co., Japan). The primary antibodies used were rabbit anti-NCC (1:100, #AB3553, Millipore), rabbit anti-phosphorylated NCC (threonine 53, 1:1000, #p1311-53, Phospho Solution). Anti-rabbit antibody with Alexa Fluor 568 (1:300, #A11011, Thermo Fisher Scientific) and anti-rabbit antibody with Alexa Fluor 488 (1:300, #A11034, Thermo Fisher Scientific) was used as the secondary antibody.

**Immunoblotting**. Proteins were extracted from whole kidneys, homogenized in lysis buffer, and whole lysate and membrane fraction were collected separately. The whole lysate and the membrane fraction were prepared in sodium dodecyl sulfate sample buffer (Cosmo Bio Co., Ltd., Tokyo, Japan). Protein concentrations were measured using the Lowry method. The extracted proteins were stored at −80 °C until immunoblotting was performed. A total of 20 μg of protein was separated using 8% non-SDS–PAGE or 10% SDS–PAGE, followed by immunoblotting. The membranes were blocked with Blocking One (Nacalai Tesque, Kyoto, Japan). The membrane and primary antibodies were incubated in Tris-buffered saline overnight at 4 °C. Anti-rabbit IgG, HRP-linked antibody (Cell Signaling, Danvers, MA), and Chemi-Lumi One (Nacalai Tesque, Kyoto, Japan) were used for detection. The

ChemiDoc XRS Plus imaging system (Bio-Rad, Hercules, CA) was used to create images of the blots. Image J software (National Institutes of Health, Bethesda, MD, USA) was used to quantify the bands. The primary antibodies used were rabbit anti-SPAK (1:500, #AB79045, Abcam), rabbit anti-phosphorylated SPAK (1:1000, #07-2273, Millipore), rabbit anti-NCC (1:1000, #AB3553, Millipore), rabbit anti-phosphorylated NCC (threonine 53, 1:1000, #p1311-53, Phospho Solution), rabbit anti-ENaCα (1:1000, #SPC-403, Stresmark Biosciences, Inc.), and rabbit anti-angiotensinogen antibody (1:1000, #AB213705, Abcam). Anti-rabbit IgG HRP antibody (1:5000, #7074, Cell Signaling Technology) was used as the secondary antibody.

**Statistics and reproducibility**. All data are presented as the mean ± SEM, and $p$-values < 0.05 were considered statistically significant. Two-tailed Student's $t$-test for experiments with two groups and the Tukey–Kramer method or two-way ANOVA for experiments including ≥3 groups were used for analysis as appropriate. Pearson's correlation coefficients were used to evaluate the relationship between salt intake and nocturnal polyuria index. All analyses were performed using JMP (SAS Institute, Cary, NC, USA) or GraphPad Prism 8.0 (GraphPad Software, SD, CA, USA).

**Reporting summary**. Further information on research design is available in the Nature Research Reporting Summary linked to this article.

## Data availability
The source data for the graphs in the main figures are included in Supplementary Data 1. Unedited blot images are provided in Supplementary Fig. 6. Other original data that support the findings of this study are available on reasonable request to the corresponding authors (H.K.).

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

## Acknowledgements

We would like to express our deepest gratitude to Mutsumi Tsuchiya and Atsuko Yasumoto for their technical assistance with experiments and to Taisuke Furusho, Motoko Chiga, Eisei Sohara, and Shinichi Uchida for their technical assistance with immunoblotting. We thank Hikari Takeshita, Yoichi Nozato, and Koichi Yamamoto for their technical assistance in measuring blood pressure. We would also like to thank Editage for English editing. This work was supported by a Grant-in-Aid for Scientific Research from the Japan Society for the Promotion of Science (JSPS) (Grant number: JP19K18582).

## Author contributions

All authors have read and agreed to the published version of the manuscript. Y.S.: Methodology, investigation, data curation, writing—original draft. H.K.: Conceptualization, supervision, writing— review and editing. K.T.: Methodology, investigation. T.I.: Investigation. S.K.: Investigation. K.O.: Investigation. Y.I.: Investigation. N.U.: Investigation. S.F.: Conceptualization, Supervision. R.I.: Investigation. H.N.: Methodology. N.N.: Conceptualization, supervision.

## Competing interests

The authors declare no competing interests.
