## [Peer Review File · Communications Biology]

Dietary salt with nitric oxide deficiency induces nocturnal polyuria in mice via hyperactivation of intrarenal angiotensin II-SPAK-NCC pathwayReviewers' comments:

Reviewer #1 (Remarks to the Author):

The authors should be commended for a well-constructed, clear manuscript. They report a new model for nocturnal polydipsia and identify inappropriately activated NCC as a critical mechanism in this model driven by the angII-SPAK pathway. The work is novel and will be an important addition to the literature.

A limitation of the manuscript is a lack of reporting on the time of day of the drug treatments, SBP measurements, molecular sampling (NCCp/ENaC). This is important because the authors are interested specifically in nocturnal polydipsia or inactive phase polydipsia. Their 2 hit model exhibits increased urine production during the active phase and so it is important for the reader to know if the drug administration/SBP and molecular measurements align with this. This and other several points for attention throughout the manuscript are raised below.

1. Supp Fig 1 "Special" – is vague, details of the type of paper would be helpful to the reader here
2. Line 66. The use of nocturnal polyuria is confusing. It would help the reader if a clear statement about mice being nocturnal was made here. Therefore, the diurnal polyuria index is the mouse equivalent of nocturnal polyuria in humans. The switch back to using "nocturnal polyuria" for the mouse model is confusing because that's not what the model is doing, in mice it's a diurnal polyuria. The use of the term "inactive phase" polyuria might be a way around this.
3. Line 72. Is this not just the same thing as an increase in DPI? (increase in inactive phase urine volume)
4. Figure 1d – how was DPI calculated (average over 3 days?), how many hours were included for inactive period/how many over the active period, this should be included in the figure legends/text.
5. Figure 1e – For the immunohistochemistry figures it would be helpful to include arrows to point out areas of glomerulosclerosis, tubular atrophy and interstitial fibrosis the authors refer to in the text. Immuno micrographs throughout need scalebars.
6. Line 78. The argument for the focus on NO is confusing/not clear in the animal studies section. This is the first mention of circadian rhythms in the manuscript and there has been no link to altered circadian genes/clocks and "inactive phase" polyuria. The argument that restoring the circadian rhythm with NO donor administration shows that NO-production directly triggers age-related conditions does not follow.
7. Supp Figure 2 and line 83. Supp 2 Does not show NOx level vs nocturnal polyuria, it shows salt vs polyuria?
8. Figure 2c&d. Is there a reason why the authors don't compare all 4 groups DPI using a two-way ANOVA? It looks like L-NAME by itself causes an increase in DPI.
9. Fig 2 The dose and how L-Name was administered should be in the figure legends, The method for measuring sBP should also be in the figure legends and time of day this was performed
10. Supp Figure 3 – Drink 1-3 is uninformative, what time blocks do these represent? With ZT used and defined preferably
11. When was SBP measured – inactive or active phase? This is critical because the mice may have elevated inactive phase blood pressure which would be masked if only measured during the active phase. Again put in figure legend/text.
12. Renal alpha ENaC subunit was estimated by immunoblots, but it wasn't easy to find that it was alpha ENaC rather than any other subunit on the figures and throughout the manuscript. Why was alpha ENaC selected over beta/gamma?
13. NCC exhibits rapid phosphorylation changes and shows time of day variability in phosphorylation levels (PMID: 26953322 and PMID: 23044422), this is directly relevant to the authors hypothesis that inappropriate NCCp is involved in polyuria. The title for Figure 3 suggests this is active phase NCC but this isn't obvious from the figure legend and text time of day for the NCCp sampling should considered and stated (in fig legends and main text).
14. NCC is phosphorylated at several key residues, pT53 is the phosphoprotein analysed here, this should be stated in the fig legends for easy reference for the reader.
15. Why was HCTZ injected s.c. rather than in the drinking water like the other treatments? (amiloride, L-NAME). This makes it difficult to compare with the amiloride treatment which would presumably be mainly taken up in the drinking water during the active period. What time of day

was the HCTZ administered? Was it a single s.c. injection of HCTZ followed by the experimental collections?

16. Line 197, "the" molecular mechanism underlying nocturnal polyuria is a strong conclusion that I'm not sure is warranted here. This is one model of nocturnal polyuria, the authors stated in the introduction that its pathogenesis is complex, with multiple co-morbidities underlying the condition. It is therefore, to my mind, unlikely that this is the only mechanism underlying nocturnal polyuria in humans.

Reviewer #2 (Remarks to the Author):

In this manuscript, the authors have created a novel mouse model of nocturia. This is an interesting study and will be of use to the community investigating nocturia. The findings are of interest. However, I have some comments, largely related to the methods and analysis, that would help my understanding of the manuscript. My specific comments are below:

1. On line 66 the authors note that urine volume was measured "at certain intervals". More details would be useful. What were the intervals, how often was urine collected, and were there an equal number of collections in the active and inactive period?

2. The histological analysis presented in Figures 1 and 2 is difficult to interpret. For example, how were glomerulosclerosis, tubular atrophy, and interstitial fibrosis quantified? Was there a significant difference between these parameters in the young and aged mice for example? Annotation of the figures would be useful to show the readers examples of tubular atrophy, glomerulosclerosis, and interstitial fibrosis.

3. In the clinical study, it is not clear to me how the urinary NO_x was divided into the high and low groups. What was the threshold for this segregation? In the methods, it is stated that both daytime and nighttime urine were collected but only 24hr NO_x is presented in Figure 1. Are there any differences in NO_x levels if daytime and nighttime urines are examined separately?

4. Urinary NO_x is reduced by LNAME in the NSD mice (Figure 2b). Is the same true for the mice fed a HSD? Along the same lines, was there an effect of HSD on urinary NO_x, independent of LNAME?

5. It would be interesting to examine whether the parameters measured in table 1 during the active period. For example, do the HSD LNAME mice drink more water during the active period, and could this contribute to the nocturia?

6. When was blood pressure measured? It would be useful to compare active and inactive blood pressure and their response to amlodipine. Were both reduced equally? Ideally, telemetry would be used to measure 24hr blood pressure which could be directly compared to urine flow.

7. When was NCC and ENaC expression measured? In the active or inactive period?

8. The data implicating the SPAK-NCC pathway is interesting. For clarity, it would be useful to state when the proteins in this pathway were measured, active or inactive period.

Reviewer #3 (Remarks to the Author):

In this study, the authors investigated the mechanism of nocturnal polyuria and reported that NO deficiency induced by L-NAME leads to increased Na excretion in inactive period, leading to osmotic diuresis and polyuria.

Unfortunately, the main finding that the impaired NO activity causes inappropriate NCC activation under a high-sodium diet has already been reported (PMID 32592815) and is not considered new. Also, there are many reports showing that the renin-angiotensin system activates NCC.

Nocturnal hypertension and altered circadian rhythm of blood pressure resulting from body sodium retention is closely related to nocturnal polyuria. In this study, authors failed to show blood pressure data in most of their experiments. They showed the effects of amlodipine in L-NAME+HSD model; however, the tail-cuff method used in the study may not be accurate especially in mice.

The urine volume of aged mice is almost doubled compared to that of young mice (Fig.1), indicating that the aged mice already showed impaired urine concentrating ability at baseline due to chronic renal impairment. This assumption is indeed supported by the histological analysis described in Fig.1e. It is well known that chronic kidney disease is a major cause of nocturnal polyuria.

Authors have proposed that angiotensin II mediates the inappropriate NCC activity in their model. However, if so, it is unclear why ENaC remained unaltered. What happened to other sodium transporters such as NHE3? If the activation of the renin-angiotensin system occurred only in distal convoluted tubules, authors need to provide evidence and rationale for the cell-selective activation.

In studies involving human urine samples, there is no information on how the patient was recruited, what the baseline characteristics was, and the relationship between age and NO in this population, etc.

Urine volume and urine sodium in inactive period were calculated by subtracting active period urine volume from total urine volume, instead of directly measuring these parameters. Given that the nocturnal polyuria is the main focus of the study, this constitutes another major limitation.

RESPONSE TO REVIEWER #1:

We wish to express our appreciation to the Reviewer for their insightful comments, which have helped us significantly improve the paper.

Comment 1: *Supp Fig 1 “Special” – is vague, details of the type of paper would be helpful to the reader here.*

Response: We appreciate the Reviewer’s comment on this point.

In this study, we used aVSOP to monitor urination behavior, which we assume would not be familiar with readers. In this method, the mouse is placed in the cage as shown in the figure below, and the paper roll below is automatically rolled up. The paper roll is a laminated filter paper pre-treated to turn the edge of urine stains deep purple to identify a spot of urine. With this device, we can accurately measure the amount and time of urination on consecutive days. This figure has already been used in other journals¹, so copyright issues would make it difficult to use.

We have changed the following text in the footnote in Supp Fig 1 from:

“Special roll of paper that turns purple when exposed to urine, used in the aVSOP method.”

to

“A laminated filter paper pre-treated to turn the edge of urine stains deep purple when exposed to urine, was used in the aVSOP method.”

1 Negoro, H. et al. Involvement of urinary bladder Connexin43 and the circadian clock in coordination of diurnal micturition rhythm. *Nat Commun* 3, 809 (2012).

Comment 2: *Line 66. The use of nocturnal polyuria is confusing. It would help the reader if a clear statement about mice being nocturnal was made here. Therefore, the diurnal polyuria index is the mouse equivalent of nocturnal polyuria in humans. The switch back to using “nocturnal polyuria” for the mouse model is confusing because that’s not what the model is doing, in mice it’s a diurnal polyuria. The use of the term “inactive phase” polyuria might be a way around this.*

Response: We thank the Reviewer for this comment.

We used mice to study the mechanism of nocturnal polyuria in humans. Since mice are nocturnal

animals, the human nighttime corresponds to the mouse daytime. As the reviewer points out, when describing nocturnal polyuria in mouse experiments, we need to be clear about whether we are referring to nocturnal polyuria as humans or nocturnal polyuria in mice. Therefore, we have clarified that mice are nocturnal and then explained the daytime polyuria index of mice, which is equivalent to the nocturnal polyuria index of humans.

The description found in lines 65–69 was inadequate. In Figure 1c & d and Figure 2c & d, we summed the amount of urine every 4 hours (determined via aVSOP method), and then calculated the three-day average for that time period. Next, we averaged the amount of urine in each group of six mice over that time period and graphed the results. In addition to that, DPi was calculated by dividing the daytime urine volume (8:00 a.m. to 8:00 p.m.) by the daily urine volume (24 hours) for each mouse and averaging it over three days, then averaging for the six mice in each group.

We have changed the following text from lines 65-69:

“Based on these recordings, we measured the urine volume at certain intervals and calculated the diurnal polyuria index in mice (Diurnal Polyuria index: DPi refers to the ratio of diurnal urine volume to daily urine volume), which was used as a corresponding index for nocturnal polyuria in humans. As expected, salt loading did not change DPi in young mice (Fig. 1c, 0.12 vs 0.13, n.s.)”

to

“Based on these recordings, urine volume was measured in each urine spot and the total amount of urine in each 4 hour period was used to show urination behavior. Since mice are nocturnal, we calculated the diurnal polyuria index in mice (Diurnal Polyuria index: DPi refers to the ratio of diurnal urine volume to daily urine volume), which was used as a corresponding index for nocturnal polyuria in humans. As expected, salt loading did not change DPi in young mice (Fig. 1c, 0.12 vs 0.13, n.s.)”

Comment 3: *Line 72. Is this not just the same thing as an increase in DPi? (increase in inactive phase urine volume)*

Response: We thank the Reviewer for this comment.

DPi was calculated by dividing daytime urine volume by daily urination volume. thus If the daily urine volume in the denominator is exactly the same for each group, then an increase in DPi is equivalent to an increase in urine volume during the inactive period. However, since the daily urine volume differs among the groups, although not significantly, DPi and daytime urine volume do not mean exactly the same thing. We believe it would be helpful for the reader's understanding if both were stated.

Comment 4: *Figure 1d – how was DPi calculated (average over 3 days?), how many hours were included for inactive period/how many over the active period, this should be included in the figure legends/text.*

Response: We thank the Reviewer for this comment.

As stated in the response to comment 2, DPi was calculated by dividing the daytime urine volume (8:00 a.m. to 8:00 p.m.) by the daily urine volume (24 hours) for each mouse and averaging it over three days, then averaging across the six mice in each group. The inactive period is from 8:00 a.m. to 8:00 p.m. and the active period is from 8:00 p.m. to 8:00 a.m., so both are 12 hours.

In accordance with the Reviewer's comment, we have added this to the legend of the figure 1c from:

“c. Mean 24-h urinary volume for three consecutive days in young NSD (black) and young HSD mice (blue) (19 weeks old), as measured via aVSOP. Daily urine volume, diurnal urine volume, and DPi (Diurnal polyuria index=diurnal urine volume/daily urine volume). Data are expressed as the mean ± SEM, * P<0.05 (n = 6 mice per group). Statistical analysis was performed using the two-tailed Student's t-test.”

to

“c. Mean 24-h urinary volume for three consecutive days in young NSD (black) and young HSD mice (blue) (19 weeks old), as measured via automated voided stain on paper (aVSOP). Daily urine volume, diurnal urine volume, and DPi (Diurnal polyuria index=diurnal urine volume/daily urine volume). The active period (night period) was from 8:00 p.m. to 8:00 a.m. and the inactive period (daytime period) was from 8:00 a.m. to 8:00 p.m. (both 12 h). DPi was calculated by averaging over three days. Data are expressed as the mean ± SEM, * P < 0.05 (n = 6 mice per group). Statistical analysis was performed using the two-tailed Student's t-test.”

Comment 5: *Figure 1e – For the immunohistochemistry figures it would be helpful to include arrows to point out areas of glomerulosclerosis, tubular atrophy and interstitial fibrosis the authors refer to in the text. Immuno micrographs throughout need scalebars.*

Response: We appreciate the reviewer's comment on this point.

It would indeed be helpful if glomerular sclerosis, tubular atrophy, and other areas were indicated with arrows on the histopathological images, so we have added them to the tissue, from:

to:

The legend of Fig.1e has also been changed accordingly from:

“e. Representative images of H&E histological staining (above) and Masson Trichrome staining (below) of kidney tissue from a young and aged mice (young NSD, young HSD, aged NSD, aged HSD). Original magnification: 200 ×. Scale bar, 100 μm.”

to

“e. Representative images of hematoxylin and eosin histological staining (above) and Masson Trichrome staining (below) of kidney tissue from a young and aged mice (young NSD, young HSD, aged NSD, aged HSD). Glomerulosclerosis (black arrowhead), tubular atrophy (white arrowhead), and interstitial fibrosis (black arrow). Original magnification: 200 ×. Scale bar, 100 μm.”

Scale bars have also been added to all immunofluorescence images.

to

The legend of Fig.3c has been changed accordingly from:

“c. Representative fluorescent immunostaining of renal phosphorylated NCC and total NCC for the four groups using serial sections. Original magnification: 200 ×.”

to

“c. Representative fluorescent immunostaining of renal phosphorylated NCC (pT53) and total NCC for the four groups using serial sections. Original magnification: 200 ×. Scale bar, 100 μm.”

Comment 6: *Line 78. The argument for the focus on NO is confusing/not clear in the animal studies section. This is the first mention of circadian rhythms in the manuscript and there has been no link to altered circadian genes/clocks and “inactive phase” polyuria. The argument that restoring the circadian rhythm with NO donor administration shows that NO-production directly triggers age-related conditions does not follow.*

Response: We wish to thank the Reviewer for this point, and after consideration we agree with you. This argument of circadian rhythms and NO donor administration would be confusing to readers. Therefore, we have eliminated this part of manuscript to avoid misunderstanding.

Lines 78-81, following sentences and reference #25 were deleted.

“In animal studies, reduced NO levels were shown to alter the circadian rhythms of PER2 and Period clock gene expression as well as the blood pressure rhythm. Further, circadian rhythm was restored through NO donor administration, suggesting that reduced NO production directly

triggered age-related conditions such as cardiovascular disorders.²⁵

Comment 7: *Supp Figure 2 and line 83. Supp 2 Does not show NOx level vs nocturnal polyuria, it shows salt vs polyuria?*

Response: We wish to thank the Reviewer for pointing this out, we have now changed the text from line 84:

“No association was found between nocturnal polyuria and NOx level (Supplementary Fig. 2)”

to

“No association was found between dietary salt intake and nocturnal polyuria (Supplementary Fig. 3)

Comment 8: *Figure 2c&d. Is there a reason why the authors don't compare all 4 groups DPi using a two-way ANOVA? It looks like L-NAME by itself causes an increase in DPi.*

Response: We appreciate the reviewer's comment.

We have since compared all four groups using two-way ANOVA and found a significant difference ($P = 0.038$). L-NAME by itself caused a significant increase in DPi ($P < 0.001$), and L-NAME+HSD further increased DPi ($P = 0.0034$).

Thus, the following plot and legend were added as Supplementary Figure 4.

Subsequent supplementary figures have been renumbered sequentially.

Supplementary figure 4. DPi in each group. Data are expressed as the mean \pm SEM, * $P < 0.05$ (NSD, HSD, L-NAME, L-NAME+HSD: $n=6$). Statistical analysis was performed using the two-way ANOVA or the Tukey-Kramer method.

Accordingly, the following underlined sentence has been inserted in the text (line 101–103).

Significant differences in the increase of DPi after high salt loading were observed between the groups treated with and without L-NAME (two-way ANOVA, $p < 0.05$) (Supplementary Fig. 4).”

Comment 9: *Fig 2 The dose and how L-Name was administered should be in the figure legends, The method for measuring sBP should also be in the figure legends and time of day this was performed.*

Response: We appreciate the reviewer's comment, and have now inserted the following underlined sentences in the legend of Figure 2d and 2f.

“Fig.2d. Mean 24 h urinary volume for three consecutive days in L-NAME (black) and L-NAME+HSD mice (red) (19 weeks old). Daily urine volume, diurnal urine volume, and DPI under L-NAME administration (5 mg/dL in drinking water).”

“Fig.2f. Effect of amlodipine (Ca channel blocker) on sBP (systolic blood pressure). sBP was evaluated in inactive period (10 a.m.) by tail-cuff method. Data are expressed as the mean \pm SEM, * P < 0.05 (sBP: n = 5 mice per group)”

Comment 10: *Supp Figure 3 – Drink 1-3 is uninformative, what time blocks do these represent? With ZT used and defined preferably.*

Response: We appreciate the reviewer's comment on this point.

In our experiments, diurnal rhythm started at 8 a.m. Thus, in Supp Figure 5, Movement 1, Food Intake 1, Drink 1 was between ZT0–ZT4 (from 8 a.m to 0 p.m.), Movement 2, Food Intake 2, Drink 2 was between ZT4–ZT8 (from 0 p.m to 4 p.m.), and Movement 3, Food Intake 3, Drink 3 was between ZT8–ZT12 (from 4 p.m to 8 p.m.).

We have thus changed the legend of Supplementary Figure 5 from:

“Salt loading and L-NAME administration do not alter behavioral patterns. The inactive period was divided into three equal parts (1, 2, 3),”

to

“Salt loading and L-NAME administration do not alter behavioral patterns.

The inactive period was divided into three equal parts (1: ZT0–ZT4, 2: ZT4–ZT8, 3: ZT8–ZT12),”

Comment 11: *When was SBP measured – inactive or active phase? This is critical because the mice may have elevated inactive phase blood pressure which would be masked if only measured during the active phase. Again put in figure legend/text.*

Response: We appreciate the reviewer's comment on this point.

We examined blood pressure in the inactive phase (10 a.m.–10 p.m.). We have revised Fig 2f to include data on blood pressure in both of active and inactive phases, from:

to:

We have changed the legend of Figure 2f from:

“Effect of amlodipine (Ca channel blocker) on sBP (systolic blood pressure). Data are expressed as the mean \pm SEM, * $P < 0.05$ (sBP: $n = 6$ mice per group)”

to

“Effect of amlodipine (Ca channel blocker) on sBP (systolic blood pressure). sBP was evaluated in active (10 p.m.) and inactive period (10 a.m.) by tail-cuff method. Data are expressed as the mean \pm SEM, * $P < 0.05$ (sBP: $n = 5$ mice per group)”

We have also changed systolic blood pressure in Table 1 from:

	NSD	HSD	L-NAME	L-NAME+HSD
Systolic Blood Pressure (mmHg)	104 \pm 2.23	103 \pm 3.01	107 \pm 1.32	122 \pm 3.50*

to

	NSD	HSD	L-NAME	L-NAME+HSD
Systolic Blood Pressure (mmHg)				
active phase	107 \pm 1.33	110 \pm 2.76	112 \pm 4.67	119 \pm 2.50*

inactive phase

96±2.69

99±4.19

103±4.28

114±3.60*

Comment 12: Renal alpha ENaC subunit was estimated by immunoblots, but it wasn't easy to find that it was alpha ENaC rather than any other subunit on the figures and throughout the manuscript. Why was alpha ENaC selected over beta/gamma?

Response: We thank the Reviewer for this comment.

Based on this, we believe it would be helpful for the readers to add ENaC β and ENaC γ to the immunoblots presented. Therefore, we have added these into Fig 3b, from:

to

We changed the legend of Figure 3b from:

“b. Representative immunoblotting (above) and quantitative analysis (below) of renal phosphorylated NCC, NCC, ENaC and ACTN for the four groups. Data are expressed as the mean \pm SEM.*P<0.05 (n=6 mice per group). Statistical analysis was performed using two-way ANOVA”

to

“b. Representative immunoblotting (above) and quantitative analysis (below) of renal phosphorylated NCC(p53), NCC, ENaC α , ENaC β , ENaC γ , and ACTN in the active phase for the four groups. Data are expressed as the mean \pm SEM.* P < 0.05 (n = 6 mice per group). Statistical analysis was performed using two-way ANOVA”

We also changed line 142 from:

“ENaC expression was not altered after salt loading, regardless of L-NAME.”

to

ENaC α , ENaC β and ENaC γ expression was not altered after salt loading, regardless of L-NAME.

Comment 13: *NCC exhibits rapid phosphorylation changes and shows time of day variability in phosphorylation levels (PMID: 26953322 and PMID: 23044422), this is directly relevant to the authors hypothesis that inappropriate NCCp is involved in polyuria. The title for Figure 3 suggests this is active phase NCC but this isn't obvious from the figure legend and text time of day for the NCCp sampling should considered and stated (in fig legends and main text).*

Response: We thank the Reviewer for this comment.

As pointed out by the Reviewer, Fig.3 is an NCC in the active phase. Therefore, we changed the figure legend from:

“b. Representative immunoblotting (above) and quantitative analysis (below) of renal phosphorylated NCC, NCC, ENaC, and ACTN for the four groups.”

to

“**b.** Representative immunoblotting (above) and quantitative analysis (below) of renal phosphorylated NCC (pT53), NCC, ENaC α , ENaC β , ENaC γ , and ACTN in the active phase for the four groups.”

We also changed the text from line 135 from:

“To investigate the molecular mechanisms underlying changes in urinary sodium excretion, we evaluated the expression of sodium chloride co-transporter (NCC) and epithelial sodium channel (ENaC), which are the main regulators of the sodium balance, in the distal tubule and collecting duct, respectively.”

to

“To investigate the molecular mechanisms underlying changes in urinary sodium excretion, we evaluated the expression of sodium chloride co-transporter (NCC) and epithelial sodium channel (ENaC) in the active phase, which are the main regulators of the sodium balance, in the distal tubule and collecting duct, respectively.”

Comment 14: *NCC is phosphorylated at several key residues, pT53 is the phosphoprotein analysed here, this should be stated in the fig legends for easy reference for the reader.*

Response: We thank the Reviewer for this comment.

Information of phosphorylated residues should be stated, thus the following underlined sentences have been inserted in the legend of Figure 3.

“**Fig.3b.** Representative immunoblotting (above) and quantitative analysis (below) of renal phosphorylated NCC (pT53), NCC, ENaC, and ACTN for the four groups. Data are expressed as the mean \pm SEM.*P < 0.05 (n = 6 mice per group). Statistical analysis was performed using two-

way ANOVA c. Representative fluorescent immunostaining of renal phosphorylated NCC (pT53) and total NCC for the four groups using serial sections. Original magnification: 200 ×.

Comment 15: *Why was HCTZ injected s.c. rather than in the drinking water like the other treatments? (amiloride, L-NAME). This makes it difficult to compare with the amiloride treatment which would presumably be mainly taken up in the drinking water during the active period. What time of day was the HCTZ administered? Was it a single s.c. injection of HCTZ followed by the experimental collections?*

Response: We thank the Reviewer for this comment.

We followed the method of previous reports on L-NAME and amiloride administration, as they are often mixed in drinking water for experiments, and HCTZ is often administered by subcutaneous injection. As the reviewer pointed out, the different routes of administration make it difficult to compare. Therefore, for diuretics such as HCTZ and amiloride, we conducted the experiment using subcutaneous injection. Therefore, we have changed Fig. 4c. from:

to:

We changed the text in line 161 from:

“we administered amiloride (5 mg/kg/day in drinking water),”

to

we administered amiloride (1.45 mg/kg/day subcutaneously),

We also changed the text in line 314–317 from:

“L-NAME (5 mg/dl, N5751, Sigma-Aldrich, St. Louis, MO), amiloride (5 mg/kg/day, A2599, Tokyo Chemical Industry, Japan), and amlodipine (6.7 mg/kg/day, A2353, Tokyo Chemical Industry, Japan) were dissolved in drinking water. Hydrochlorothiazide (20 mg/kg/day, H4759, Sigma-

Aldrich) was administered via subcutaneous injection.”

to:

L-NAME (5 mg/dl, N5751, Sigma-Aldrich, St. Louis, MO) and amlodipine (6.7 mg/kg/day, A2353, Tokyo Chemical Industry, Japan) were dissolved in drinking water. Hydrochlorothiazide (20 mg/kg/day, H4759, Sigma-Aldrich) and amiloride (1.45 mg/kg/day, A2599, Tokyo Chemical Industry, Japan) was administered via subcutaneous injection.

Comment 16: *Line 197, “the” molecular mechanism underlying nocturnal polyuria is a strong conclusion that I’m not sure is warranted here. This is one model of nocturnal polyuria, the authors stated in the introduction that its pathogenesis is complex, with multiple co-morbidities underlying the condition. It is therefore, to my mind, unlikely that this is the only mechanism underlying nocturnal polyuria in humans.*

Response: We thank the Reviewer for this comment.

As you state, our model is just one model of nocturnal polyuria. “The” molecular mechanism underlying nocturnal polyuria is therefore a strong conclusion. Thus, we have changed the following text in line 198 from:

“we were able to uncover the molecular mechanism underlying nocturnal polyuria.”

to

“we were able to uncover one of the molecular mechanisms underlying nocturnal polyuria.”

RESPONSE TO REVIEWER #2:

We wish to express our appreciation to the Reviewer for their insightful comments, which have helped us significantly improve the paper.

Comment 1: *On line 66 the authors note that urine volume was measured “at certain intervals”. More details would be useful. What were the intervals, how often was urine collected, and were there an equal number of collections in the active and inactive period?*

Response: We thank the Reviewer for this comment.

In this study, we used the aVSOP method to monitor urination behavior. In this method, the mouse is placed in the cage as shown in the figure below, and the paper roll below is automatically rolled up. The paper roll is a laminated filter paper pre-treated to turn the edge of urine stains deep purple to enable identification of a spot of urine. With this device, we can accurately measure the amount and duration of mice urination on consecutive days.

We collected data for three consecutive days and summed the amount of urine measured every four hours, and then averaged the amount of urine across each group of six mice every four hours. Fig. 1c and 2c represent the results for these three days.

Therefore, we did not actually collect the urine, but calculated the amount.
we have changed the following text in lines 65–69:

“Based on these recordings, we measured the urine volume at certain intervals and calculated the diurnal polyuria index in mice (Diurnal Polyuria index: DPi refers to the ratio of diurnal urine volume to daily urine volume), which was used as a corresponding index for nocturnal polyuria in humans. As expected, salt loading did not change DPi in young mice (Fig. 1c, 0.12 vs 0.13, n.s.)”

to

“Urine volume was measured in each urine spot and the total amount of urine in each 4 h period was used to show urination behavior. Then, we calculated the diurnal polyuria index in mice (Diurnal Polyuria index: DPi refers to the ratio of diurnal urine volume to daily urine volume), which was used as a corresponding index for nocturnal polyuria in humans. As expected, salt loading did not change DPi in young mice (Fig. 1c, 0.12 vs 0.13, n.s.)”

Comment 2: *The histological analysis presented in Figures 1 and 2 is difficult to interpret. For*

example, how were glomerulosclerosis, tubular atrophy, and interstitial fibrosis quantified? Was there a significant difference between these parameters in the young and aged mice for example? Annotation of the figures would be useful to show the readers examples of tubular atrophy, glomerulosclerosis, and interstitial fibrosis.

Response: We thank the Reviewer for this comment.

As the Reviewer pointed out, it would be useful for the readers to annotate the figures with examples of glomerulosclerosis, tubular atrophy and interstitial fibrosis, and further quantify them to compare.

Thus, we have added annotation to Fig. 1e:

from:

to

The legend of Fig.1e has been changed accordingly, from:

“e. Representative images of H&E histological staining (above) and Masson Trichrome staining (below) of kidney tissue from a young and aged mice (young NSD, young HSD, aged NSD, aged HSD). Original magnification: 200 ×. Scale bar, 100 μm.”

to

“e. Representative images of hematoxylin and eosin histological staining (above) and Masson Trichrome staining (below) of kidney tissue from a young and aged mice (young NSD, young HSD, aged NSD, aged HSD). Glomerulosclerosis (black arrowhead), tubular atrophy (white arrowhead), and interstitial fibrosis (black arrow) Original magnification: 200 ×. Scale bar, 100 μm.”

We have quantified alteration of the kidney tissue and added a new figure as Supplementary Figure 2.

Supplementary Figure 2. Histological quantification of the effect of salt loading on the kidney in a young mice and aged mice. a. Glomerulosclerosis score, b. Tubular injury score, c. Interstitial fibrosis score. Data are expressed as the mean ± SEM, * P < 0.05 (n = 5 mice per group). Statistical analysis was performed using a two-tailed Student's t-test.

We have inserted the following text in Method in line 355-356 and added the references 62 and 63.

Glomerulosclerosis, tubular injury, and interstitial fibrosis were assessed based on scores from previous reports^{62,63}.

Comment 3: *In the clinical study, it is not clear to me how the urinary NOx was divided into the high and low groups. What was the threshold for this segregation? In the methods, it is stated that both daytime and nighttime urine were collected but only 24hr NOx is presented in Figure 1. Are there any differences in NOx levels if daytime and nighttime urines are examined separately?*

Response: We thank the Reviewer for this comment.

The high and low NOx groups were divided by median values because there was no clear normal range. The threshold of NOx value was 0.25 μmol/Cr mg.

We collected daytime urine and nighttime urine separately, but to measure NOx concentrations during the day, we mixed the two in the ratio of daytime urine volume to nighttime urine volume. We did not measure NOx separately for day and night.

We have added this table as Supplemental Table 1.

Supplementary Table 1 Characteristics of 27 patients. Data are expressed as median (range), or number.

	All patients (n = 27)	Low NO (n = 13)	High NO (n = 14)
Age, yr	61 (42–79)	57 (44–74)	65 (42–79)
Sex (M/F), no.	10/17	7/6	3/11
Nocturnal Polyuria Index	0.39 (0.16–0.65)	0.40 (0.24–0.59)	0.39 (0.16–0.65)
serum Cr, mg/dL	0.71 (0.47–1.20)	0.77 (0.53–1.20)	0.65 (0.47–1.01)
eGFR, ml/min/1.73m ²	75.4 (48.6–98.8)	73.5 (48.6–94.5)	77.1 (56.5–98.58)
NOx, μ mol/Cr mg	0.29 (0.02–0.58)	0.17 (0.02–0.25)	0.39 (0.26–0.58)
Urinary salt excretion, g	6.8 (2.8–15.4)	8.2 (4.0–15.4)	5.5 (2.8–8.8)

Unfortunately, there was only a small number of urine sample left, so we evaluated NOx value in daytime and nighttime separately in seven subjects.

	All patients (n = 7)
Age, yr	61 (42–79)
Sex (M/F), no.	4/3
Daytime NOx, μ mol	154 (89–268)
Nighttime NOx, μ mol	117 (46–241)
Daily NOx, μ mol/	271 (186–377)
Daytime NOx, μ mol/Cr mg	0.40 (0.17–0.80)
Nighttime NOx, μ mol/Cr mg	0.31 (0.19–0.41)
Daily NOx, μ mol/Cr mg	0.37 (0.18–0.68)
daytime/nighttime NOx ratio	1.23 (0.16–0.65)
Nocturnal Polyuria Index	0.43 (0.16–0.65)
serum Cr, mg/dL	0.72 (0.51–0.94)

Although an interesting proposition, in these small groups we did not find an association between urinary NOx excretion in daytime and nighttime and nocturnal polyuria.

Comment 4: *Urinary NOx is reduced by LNAME in the NSD mice (Figure 2b). Is the same true for the mice fed a HSD? Along the same lines, was there an effect of HSD on urinary NOx, independent of LNAME?*

Response: We thank the Reviewer for this comment.

Measurements of urinary NOx levels in HSD mice have been added. Urinary NOx levels showed a significant decrease after L-NAME administration in HSD mice. No difference was found between NSD and HSD mice in urinary NOx level, suggesting that HSD alone had no effect on NOx level, and that L-NAME lowered NOx level in both NSD and HSD.

Thus, we have changed Fig. 2b from:

to

The legend of Fig.2b has been changed accordingly, from:

“b. Urinary NOx concentration adjusted by urinary creatinine level in NSD and L-NAME groups. Data are expressed as the mean \pm SEM, * $P < 0.05$ (n =5). Statistical analysis was performed using the two-tailed Student's *t*-test.”

to:

“b. Urinary NOx concentration adjusted by urinary creatinine level in NSD, HSD, L-NAME and L-NAME+HSD groups. Data are expressed as the mean \pm SEM, * $P < 0.05$ (n =5). Statistical analysis was performed using the Tukey-Kramer method.”

Comment 5: *It would be interesting to examine whether the parameters measured in table 1 during the active period. For example, do the HSD LNAME mice drink more water during the active period, and could this contribute to the nocturia?*

Response: We thank the Reviewer for this comment.

In accordance with Reviewer 2's comment, we have added food intake, water intake and systolic

blood pressure to Table 1, from:

	NSD	HSD	L-NAME	L-NAME+HSD
Body Weight (g)	29.6 ± 0.68	29.8 ± 0.67	29.2 ± 0.33	29.1 ± 0.82
Food Intake (g)	3.85 ± 0.27	3.53 ± 0.11	3.82 ± 0.17	4.20 ± 0.14
Water Intake (g)	5.02 ± 0.11	4.82 ± 0.10	4.93 ± 0.16	4.75 ± 0.19
Systolic Blood Pressure (mmHg)	104 ± 2.23	103 ± 3.01	107 ± 1.32	122 ± 3.50*
Serum Na (mEq/L)	149 ± 2.99	148 ± 0.93	150 ± 0.98	152 ± 1.85
Serum Cr (mg/dL)	0.13 ± 0.01	0.13 ± 0.01	0.16 ± 0.01*	0.17 ± 0.02*

to

	NSD	HSD	L-NAME	L-NAME+HSD
Body Weight (g)	29.6 ± 0.68	29.8 ± 0.67	29.2 ± 0.33	29.1 ± 0.82
Food Intake (g)				
active phase	3.58 ± 0.32	3.22 ± 0.26	3.32 ± 0.40	3.50 ± 0.40
inactive phase	0.52 ± 0.19	0.66 ± 0.29	0.58 ± 0.18	0.74 ± 0.19
total	4.10 ± 0.45	3.88 ± 0.37	3.90 ± 0.27	4.24 ± 0.27
Water Intake (g)				
active phase	4.42 ± 0.22	4.30 ± 0.42	4.34 ± 0.49	4.66 ± 0.39
inactive phase	0.60 ± 0.12	0.72 ± 0.19	0.76 ± 0.18	0.66 ± 0.09
total	5.02 ± 0.28	5.02 ± 0.28	5.10 ± 0.48	5.32 ± 0.36
Systolic Blood Pressure (mmHg)				
active phase	107 ± 1.33	110 ± 2.76	112 ± 4.67	119 ± 2.50*
inactive phase	96 ± 2.69	99 ± 4.19	103 ± 4.28	114 ± 3.60*
Serum Na (mEq/L)	149 ± 2.99	148 ± 0.93	150 ± 0.98	152 ± 1.85
Serum Cr (mg/dL)	0.13 ± 0.01	0.13 ± 0.01	0.16 ± 0.01*	0.17 ± 0.02*

Comment 6: *When was blood pressure measured? It would be useful to compare active and inactive blood pressure and their response to amlodipine. Were both reduced equally? Ideally, telemetry would be used to measure 24hr blood pressure which could be directly compared to urine flow.*

Response: We thank the Reviewer for this comment.

Blood pressure was measured in the inactive phase (10 a.m). Blood pressure measurements taken in the inactive phase were added to Fig.2f, from:

to:

The legend of Fig.2f has been changed accordingly, from:

“f. Effect of amlodipine (Ca channel blocker) on sBP (systolic blood pressure). Data are expressed as the mean \pm SEM, * $P < 0.05$ (sBP: $n = 6$ mice per group)”

to:

f. Effect of amlodipine (Ca channel blocker) on sBP (systolic blood pressure). sBP was evaluated in active (10 p.m.) and inactive period (10 a.m.) by tail-cuff method. Data are expressed as the mean \pm SEM, * $P < 0.05$ (sBP: $n = 5$ mice per group)

Comment 7: *When was NCC and ENaC expression measured? In the active or inactive period?*

Response: We thank the Reviewer for this comment.

We euthanized the mice during the active phase and examined the protein expression of NCC and ENaC. Thus, the protein expression was evaluated in the active phase.

Figure 3 was altered from:

to

The legend of Fig.3b has been changed accordingly, from:

“b. Representative immunoblotting (above) and quantitative analysis (below) of renal phosphorylated NCC, NCC, ENaC, and ACTN for the four groups.”

to

Representative immunoblotting (above) and quantitative analysis (below) of renal phosphorylated NCC (pT53), NCC, ENaCα, ENaCβ, ENaCγ, and ACTN in the active phase for the four groups.

Comment 8: The data implicating the SPAK-NCC pathway is interesting. For clarity, it would be useful to state when the proteins in this pathway were measured, active or inactive period.

Response: We thank the Reviewer for this comment.

Mice were euthanized in the active phase to assess levels of SPAK and angiotensinogen in the kidney and aldosterone in the serum.

The legend of Fig.5a and 5c has been changed accordingly, from:

“a. Representative immunoblotting (above) and quantitative analysis (below) of renal phosphorylated SPAK and total SPAK in the four groups.”

to:

“a. Representative immunoblotting (above) and quantitative analysis (below) of renal phosphorylated STE20/SPS1-related proline–alanine-rich protein kinase (SPAK) and total SPAK in the active phase in the four groups.”

from:

“c. Representative immunoblotting (above) and quantitative analysis (below) of renal AGT (angiotensinogen) among the four groups.”

to

“c. Representative immunoblotting (above) and quantitative analysis (below) of renal AGT (angiotensinogen) in the active phase among the four groups.”

We have also changed text from (Line 311)

“Mice were then sacrificed by inhalation of isoflurane,”

to

“Mice were then sacrificed in the active phase by inhalation of isoflurane,”

RESPONSE TO REVIEWER #3:

We wish to express our appreciation to the Reviewer for their insightful comments, which have helped us significantly improve the paper.

Comment 1: *Nocturnal hypertension and altered circadian rhythm of blood pressure resulting from body sodium retention is closely related to nocturnal polyuria. In this study, authors failed to show blood pressure data in most of their experiments. They showed the effects of amlodipine in L-NAME+HSD model; however, the tail-cuff method used in the study may not be accurate especially in mice.*

Response: We appreciate the Reviewer's comment.

We agree that nocturnal hypertension and altered circadian rhythm of blood pressure resulting from bodily sodium retention is closely related to nocturnal polyuria. Several clinical studies have suggested the association between hypertension and nocturnal polyuria², and one recent study showed that nocturnal polyuria is associated with nocturnal hypertension³. However, studies on the association between nocturnal polyuria and nocturnal hypertension are still limited and no studies have revealed a causal relationship. This is a field of interest that needs more clinical research in the future. In clinical settings, lowering blood pressure via anti-hypertensive drugs in patients with hypertension and nocturnal polyuria doesn't often improve nocturnal polyuria. In our study, amlodipine also did not improve nocturnal polyuria, despite lowering blood pressure. Together, these results suggest that some hypertensions and nocturnal polyuria share a common causative factor, and this common factor is the accumulation of salt in the body during the daytime. We found that daytime salt retention is related to inappropriate activation of the RAS-NCC pathway in the kidney, leading to nocturnal polyuria. In this study, we chose to focus on the RAS-NCC pathway intrarenal pathway, and so we will leave detailed study on blood pressure to other researchers. As for the assessment of blood pressure, ideally it would be better to measure blood pressure for 24 h using telemetry, but since the tail-cuff method is an established method for blood pressure measurement, we used the tail-cuff method in this study. This is a limitation of our study. The following sentence have been inserted in line 288–291:

“Third, we used the tail-cuff method to evaluate blood pressure, but measurement of blood pressure for 24 h using telemetry would be better to assess the relationship between blood pressure and nocturnal polyuria in detail.”

ref.

2. Yokoyama O, et al., Nocturnal Polyuria and Hypertension in Patients with Lifestyle Related Diseases and Overactive Bladder. *J Urol.* 2017 Feb;197(2):423-431.
3. Takayama M, et al., Three-year safety, efficacy and persistence data following the daily use of mirabegron for overactive bladder in the clinical setting: A Japanese post-marketing surveillance study. *Low Urin Tract Symptoms.* 2019 Apr;11(2): O98-O102

Comment 2: *The urine volume of aged mice is almost doubled compared to that of young mice*

(Fig. 1), indicating that the aged mice already showed impaired urine concentrating ability at baseline due to chronic renal impairment. This assumption is indeed supported by the histological analysis described in Fig. 1e. It is well known that chronic kidney disease is a major cause of nocturnal polyuria.

Response: We appreciate the Reviewer’s comment.

Impaired renal function is well-known to lead to a decrease urinary concentration and thus nocturnal polyuria. As the reviewer points out, baseline kidney function in aged mice seemed impaired and these mice exhibited nocturnal polyuria when compared with young mice. Based on the hypothesis that there may be other causes of nocturnal polyuria besides impaired urine concentrating ability, we conducted this animal study focusing on urinary sodium excretion. We have conducted a longitudinal clinical study in renal transplant donor patients to explore the causes of nocturnal polyuria in patients with impaired renal function.⁴ In that study, we found that the most influential cause of nocturnal polyuria was not a deficit in urinary concentration (urine osmolality), but increased nighttime sodium excretion. Of course, deficit in urinary concentration (urine osmolality) also had an impact on nocturnal polyuria. As the reviewer pointed out, decreased ability to concentrate urine is a major factor in nocturnal polyuria. In this study, we would choose to focus on inappropriate activation of the intrarenal RAS-NCC pathway and impaired sodium excretion as a new possible mechanism of nocturnal polyuria.

Cited from ref.4

Variables	Simple linear regression			Multiple linear regression			
	Coefficient	(95% CI)	P	Unstandardized coefficient	(95% CI)	Standardized coefficient	P
Age	0.10	(-0.27, 0.48)	0.59	0.27	(0.12, 0.42)	0.21	<0.01
Pre eGFR	-0.01	(-0.35, 0.32)	0.94				
ΔeGFR	-0.13	(-0.82, 0.55)	0.70				
ΔNighttime urine osmolality	-0.02	(-0.07, 0.02)	0.35	0.03	(0.01, 0.05)	0.17	<0.01
ΔNighttime salt excretion rate	0.73	(0.55, 0.91)	<0.01	0.75	(0.48, 1.03)	0.79	<0.01
ΔNighttime K excretion rate	0.69	(0.48, 0.91)	<0.01	-0.36	(-0.73, 0.01)	-0.36	0.06
ΔNighttime urea excretion rate	0.79	(0.60, 0.99)	<0.01	0.42	(-0.02, 0.86)	0.40	0.06

Table 2. Factors associated with the increase in the nighttime urine volume rate. *Pre* Preoperative, *eGFR* estimated glomerular filtration rate.

ref.

4. Takezawa K, et al., Decreased renal function increases the nighttime urine volume rate by carryover of salt excretion to the nighttime. *Sci Rep.* 2021 May 19;11(1):10587.

Comment 3: Authors have proposed that angiotensin II mediates the inappropriate NCC activity in their model. However, if so, it is unclear why ENaC remained unaltered. What happened to other sodium transporters such as NHE3? If the activation of the renin-angiotensin system occurred only in distal convoluted tubules, authors need to provide evidence and rationale for the cell-selective activation.

Response: We thank the Reviewer for this pertinent comment.

The expression of NHE3 in proximal tubules was compared among the NSD, HSD, L-NAME, and L-NAME+HSD groups. The expression levels in NSD and HSD were similar, consistent with previous reports.⁵ However, L-NAME tended to increase its expression, although this was not statistically significant. This is also in agreement with previous reports⁶. L-NAME+HSD decreased its expression slightly, although not significantly. These results indicate high salt itself doesn't alter Na reabsorption by NHE3, and that L-NAME may increase Na reabsorption. Further research is needed on regulation of salt balance by NHE3. In this paper, we focused on the Na transporters, NCC and ENaC in distal tubules and collecting ducts, so including the results of NHE3 may confuse readers. Therefore, we would like to include the fact that we did not evaluate proximal tubular transporters in the limitations. We appreciate your comments and suggestions to improve the quality of our paper.

Unedited gel

rabbit anti-NHE3(1:1000, #GTX41967, Gene Tex)

ref.

5. Torres-Pinzon DL, et al. Sex-specific adaptations to high-salt diet preserve electrolyte homeostasis with distinct sodium transporter profiles. *Am J Physiol Cell Physiol*. 2021 Nov 1;321(5):C897-C909

6. Kekuda R, et al. Constitutive nitric oxide differentially regulates Na-H and Na-glucose cotransport in intestinal epithelial cells. *Am J Physiol Gastrointest Liver Physiol*. 2008 Jun;294(6):G1369-75.

As to why activated RAS pathway had little effect on ENaC but only NCC, we do not have enough data to provide a rationale for this. There have been previous reports similar with our results⁵, implying there may be differences in such results among animal models.

Cited from ref.7

Figure 3 | Sodium-chloride cotransporter (NCC), but not epithelial sodium channel (ENaC), was activated in *N*-nitro-L-arginine methyl ester (L-NAME)-induced salt-sensitive hypertension in 8-week-old C57BL/6J mice. (a) Effects of hydrochlorothiazide (HCTZ) in the high-salt diet (HS) and HS + L-NAME groups. After 4 weeks of treatment with HS and HS + L-NAME, we measured sodium excretion (continued)

ref.

7. Wang C, et al., Low-dose L-NAME induces salt sensitivity associated with sustained increased blood volume and sodium-chloride cotransporter activity in rodents. *Kidney Int* 2020 Nov;98(5):1242-1252.

Comment 4: *In studies involving human urine samples, there is no information on how the patient was recruited, what the baseline characteristics was, and the relationship between age and NO in this population, etc.*

Response: We thank the Reviewer for this pertinent comment.

We regret that our human study lacked patient characteristics. In this study, we collected urine and frequency-volume charts from preoperative donor patients of kidney transplant. The NO group was divided into two groups by median NO concentration. Baseline characteristics are as follows:

We have added this table to Supplemental Tables and inserted text into lines 81–83.

Supplementary Table 1 Characteristics of 27 patients. Data are expressed as median (range), or number.

	All patients (n = 27)	Low NO (n = 13)	High NO (n = 14)
Age, yr	61 (42–79)	57 (44–74)	65 (42–79)
Sex (M/F), no.	10/17	7/6	3/11

serum Cr, mg/dL	0.71 (0.47–1.20)	0.77 (0.53–1.20)	0.65 (0.47–1.01)
eGFR, ml/min/1.73m ²	75.4 (48.6–98.8)	73.5 (48.6–94.5)	77.1 (56.5–98.58)
NOx, μ mol/Crmg	0.29 (0.02–0.58)	0.17 (0.02–0.25)	0.39 (0.26–0.58)
Urinary salt excretion, g	6.8 (2.8–15.4)	8.2 (4.0–15.4)	5.5 (2.8–8.8)
Nocturnal Polyuria Index	0.39 (0.16–0.65)	0.40 (0.24–0.59)	0.39 (0.16–0.65)

“In order to assess the association between nocturnal polyuria and NO production in humans, we collected 24 h urine and measured urinary NO₂/NO₃ (= NOx) levels, which reflect the amount of NO production (Supplementary Table 1).”

Accumulated evidence shows that in general, NO declines as age increases, but in this small population, no correlation was found between NOx and age ($p = 0.48$). Notably, the correlation between dietary salt intake and nocturnal polyuria, which is typically only found in the elderly, showed a significant correlation in the low NO group, which was not elderly.

Figure Correlation of age and NOx level.

Comment 5: *Urine volume and urine sodium in inactive period were calculated by subtracting active period urine volume from total urine volume, instead of directly measuring these parameters. Given that the nocturnal polyuria is the main focus of the study, this constitutes another major limitation.*

Response: We thank the Reviewer for this pertinent comment.

Since urine volume and urine sodium in the inactive phase was very small, it was very difficult to evaluate directly. As the Reviewer pointed out, this was one of the limitations. Thus, we inserted this in the discussion.

Accordingly, the following underlined sentence has been inserted in the text (Line 290–291).

“Fourth, the volume of urine during the inactive period was small, so urine volume and urinary electrolyte concentration were calculated.”

REVIEWERS' COMMENTS:

Reviewer #1 (Remarks to the Author):

I'm happy that the authors have addressed all of my concerns. However, the scale bars in all of the images are almost impossible to see, can the thickness of these and colour/contrast be considered so that they're visible for the reader.

Reviewer #2 (Remarks to the Author):

I would like to thank the authors for addressing my concerns, I have no further comments.

Reviewer #3 (Remarks to the Author):

The reviewer would like to thank the authors for addressing the comments and concerns raised. No further comments.

RESPONSE TO REVIEWER #1:

We wish to express our appreciation to the Reviewer for their insightful comment, which have helped us significantly improve the paper.

Comment 1: *The scale bars in all of the images are almost impossible to see, can the thickness of these and colour/contrast be considered so that they're visible for the reader.*

Response: We appreciate the Reviewer's comment on this point. We have made the scale bar thicker.

RESPONSE TO REVIEWER #2:

We wish to express our appreciation to the Reviewer for improving the paper.

RESPONSE TO REVIEWER #3:

We wish to express our appreciation to the Reviewer for improving the paper.